# Characterization of the Nitrogen Stable Isotope Composition ($\delta^{15}$N) of Ship-emitted NO$_x$

Zeyu Sun[1,2,8], Zheng Zong[1,3], Yang Tan[1], Chongguo Tian[1,2,7,*], Zeyu Liu[4], Fan Zhang[5], Rong Sun[1,2], Yingjun Chen[4], Jun Li[6], Gan Zhang[6]

[1]CAS Key Laboratory of Coastal Environmental Processes and Ecological Remediation, Yantai Institute of Coastal Zone Research, Chinese Academy of Sciences, Yantai, 264003, China
[2]Shandong Key Laboratory of Coastal Environmental Processes, Yantai, 264003, China
[3]Department of Civil and Environmental Engineering, Hong Kong Polytechnic University, Hong Kong, 999077, China
[4]Shanghai Key Laboratory of Atmospheric Particle Pollution and Prevention (LAP), Department of Environmental Science and Engineering, Fudan University, Shanghai, 200438, China
[5]Key Lab of Geographic Information Science of the Ministry of Education, School of Geographic Sciences, East China Normal University, Shanghai, 200241, China
[6]State Key Laboratory of Organic Geochemistry and Guangdong Key Laboratory of Environmental Protection and Resources Utilization, Guangzhou Institute of Geochemistry, Chinese Academy of Sciences, Guangzhou, 510640, China
[7]Center for Ocean Mega–Science, Chinese Academy of Sciences, Qingdao, 266071, China
[8]University of Chinese Academy of Sciences, Beijing, 100049, China

*Correspondence to*: Chongguo Tian (cgtian@yic.ac.cn)

**Abstract.** The nitrogen stable isotope composition ($\delta^{15}$N) of nitrogen oxides (NO$_x$) is a powerful indicator for source apportionment of atmospheric NO$_x$; however, $\delta^{15}$N–NO$_x$ values emitted from ships have not been reported, affecting the accuracy of source partitioning of atmospheric NO$_x$ in coastal zones with a lot of vessel activity. In addition, $\delta^{15}$N–NO$_x$ values from ship emissions could also be important for source apportionment of atmospheric nitrogen deposition in remote ocean regions. This study systemically analyzed the $\delta^{15}$N–NO$_x$ variability and main influencing factors of ship emissions. Results showed that $\delta^{15}$N–NO$_x$ values from ships, which were calculated by weighting the emission values from the main engine and auxiliary engine of the vessel, ranged from −35.8 ‰ to 2.04 ‰ with a mean ± standard deviation of −18.5 ± 10.9 ‰. The $\delta^{15}$N–NO$_x$ values increased monotonically with the ongoing tightening of emission regulations, presenting a significantly negative logarithmic relationship with NO$_x$ concentrations ($p < 0.01$). The selective catalytic reduction (SCR) system was the most important factor affecting changes in $\delta^{15}$N–NO$_x$ values, followed by the ship category, fuel types and operation states of ships. Based on the relationship between $\delta^{15}$N–NO$_x$ values and emission regulations observed in this investigation, a mass-weighted model to compute accurate assessments over time was developed and the temporal variation in $\delta^{15}$N–NO$_x$ values from ship emissions in the international merchant fleet was evaluated. These simulated $\delta^{15}$N–NO$_x$ values can be used to select suitable $\delta^{15}$N–NO$_x$ values for a more accurate assessment, including the contribution of ship-emitted exhaust to atmospheric NO$_x$ and its influence on atmospheric nitrate (NO$_3^-$) air quality and nitrogen deposition studies.

**Graphical abstracts**

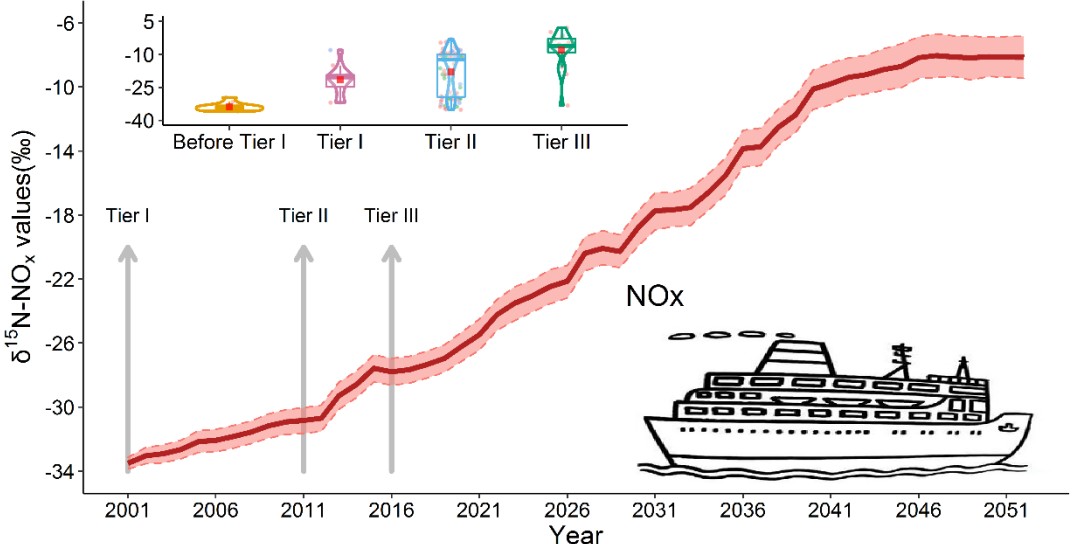

# 1 Introduction

Due to its detrimental impact on the environment, ecology, and public health, anthropogenic emissions of nitrogen oxides ($NO_x$), including nitric oxide (NO) and nitrogen dioxide ($NO_2$), have drawn considerable attention in recent decades. Transportation exhaust (motor vehicles and ships) tends to overtake coal combustion as the most important source of $NO_x$ emissions in anthropogenic activities (Zhang et al., 2020; Jin et al., 2021; Shi et al., 2021), contributing 48 % of total emissions at a global scale in 2014 (Huang et al., 2017). Air pollution from navigation has become a growing concern in maritime regions with increasing demand for global and regional trade activities (Johansson et al., 2017; Nunes et al., 2017). To illustrate, the $NO_x$ emissions from ships in China, a major maritime trading country, have reached 130–220 kt, accounting for approximately 34 % of national motor vehicle $NO_x$ emissions (Fu et al., 2017). International organizations and national governments have developed a series of regulations on marine traffic activities, prescribing limits for $NO_x$ emissions. For instance, the International Convention for the Prevention of Pollution from Ships (MARPOL) implemented by the International Maritime Organization (IMO) is a momentous international treaty concerning the avoidance of pollution generated by vessels during operations or from unexpected causes. To mitigate the negative effects of navigation on the environment, the IMO set stringent $NO_x$ emission regulations with a decrease of 16–22 % and 80 % in 2011 (Tier II) and 2016 (Tier III), respectively, in comparison with the year 2000 (Tier I) through MARPOL Annex VI (https://dieselnet.com/standards/inter/imo.php#other). While improving the quality of marine fuel, solutions for marine diesel engines that reduce emissions, mainly including exhaust after-treatment systems, precombustion control techniques and fuel optimizations are rapidly developing and applied to newly built ships for future, stricter requirements of $NO_x$ pollution abatement (Deng et al., 2021; Selleri et al., 2021). In addition, the statistics reveal that the world merchant fleet has ships of 2.184 billion deadweight tonnage (DWT) with an average age of 11.8 years in 2021, compared with 797 million DWT with an average age of 18.8 years in 2001 (Wei et al., 2022; Yang and Zhou, 2002). The current world merchant fleet thus comprises more newly built vessels that have

benefited from the implementation of emission reduction technologies. The rapid development of the shipping industry indicates that emission factors of $NO_x$ from ships are changing significantly.

The emission factor (EF) of a ship refers to the parameter that measures the rate or proportion of pollutants generated by the ship during its operation, usually expressed in grams per kilowatt-hour (g $(KWh)^{-1}$). The g $(KWh)^{-1}$ value indicates the number of pollutants emitted per kilowatt-hour of energy
consumed. A lower EF value indicates lower emissions of pollutants per unit of energy consumed, indicating a cleaner and more environmentally friendly combustion process. Emission factors are important basic data for compiling emission inventories, on which adequate and reliable information is crucial to work effectively on controlling emissions and associated health impacts for policy makers. Early assessment, not providing high precision spatiotemporal variability in ship emissions because of
limited data availability, showed that about 70 % of ship emissions occurred within 400 km of the coast based on emission factors and proxies highly pertinent to ship activities (e.g. fuel usage, port throughput, and seaborne trade) (Corbett and Fischbeck, 1997). A recent evaluation suggested that 60 % of total emissions happened within 20 nautical miles (Nm) of the coast on the basis of emission factors and historic ship traffic activities by detailed Automatic Identification System (AIS) data, which is able to
provide a ship emission inventory with high spatiotemporal resolution(Liu et al., 2016). Simulations of atmospheric air quality models further revealed that the intensive emissions from ships within 12 Nm from the coastline were the primary contributor to ship-related $NO_x$, accounting for approximately 70 % of the total emissions (Wang et al., 2018). These studies of ship emissions and their environmental impacts indicate that the effect of ship emissions on air quality in offshore zones is becoming increasingly
significant. Thus, it is critical to estimate ship-emitted $NO_x$ and their impact as precisely and thoroughly as feasible. Due to the variation in $NO_x$ emission factors from ships, additional methods independent of the emission factors are necessary to evaluate the impact of ship emissions.

$NO_x$ released into the atmosphere principally oxidizes to nitrate ($NO_3^-$) and nitric acid ($HNO_3$) and their nitrogen stable isotope composition ($\delta^{15}N$, i.e. $^{15}N/^{14}N$, expressed in ‰) is a powerful method to
apportion $NO_x$ sources due to the significant differences in $\delta^{15}N$ values of $NO_x$ ($\delta^{15}N–NO_x$) from different sources (Walters et al., 2015a; Walters et al., 2015b; Zong et al., 2020a). In recent years, $\delta^{15}N$ values of $NO_3^-$ ($\delta^{15}N–NO_3^-$) in the atmosphere have been proverbially adopted in tracing sources of atmospheric $NO_x$ based on Bayesian models and $\delta^{15}N–NO_x$ values characterized for various sources (Song et al., 2019; Zong et al., 2020b; Zong et al., 2017). The considered sources were mainly coal combustion, vehicle
exhaust, biomass burning and biological soil emissions (Luo et al., 2019; Song et al., 2020; Song et al., 2019; Zong et al., 2020b; Zhu et al., 2021; Yin et al., 2022). However, ship emissions were not considered because $\delta^{15}N–NO_x$ values emitted from ships have not been reported so far. Lack of $\delta^{15}N–NO_x$ values from ships affects the accuracy of atmospheric $NO_x$ source apportionment based on $\delta^{15}N$ signals in offshore areas, especially in some ports with frequent ship activities. To address the lack of $\delta^{15}N–NO_x$
measurements associated with ship emissions, this study systemically collected $NO_x$ discharged by ships to analyze the variation in $\delta^{15}N–NO_x$ values and their possible influencing factors. Furthermore, based on the $\delta^{15}N–NO_x$ values in our findings, the temporal variation in $\delta^{15}N–NO_x$ values from ship emissions in the international merchant fleet was evaluated by developing a mass-weighted model. These insights have implications for assessing changes in $\delta^{15}N–NO_x$ values of ship emissions and its potential

 applications in source apportionment.

## 2 Methodology

### 2.1 Sampling campaign

$NO_x$ samples were collected from four types of ships between January 2020 and April 2021. These ships included 5 cargo ships (SH1–SH5), 2 fishing boats (Y1, Y2), 1 passenger ship (K1), and 1 research ship (KK1). The number of ships per category applied for $NO_x$ sample collection was determined based on previous reports that cargo ships accounted for more than 50 % of all $NO_x$ emissions from ships in China in 2014 and the fuel consumed by fishing boats accounted for 40 % of all ship fuel use in China in 2011 (Chen et al., 2017; Zhang et al., 2018; Zong et al., 2017). The technical parameters of ships utilized for sample collection are listed in Table 1. Under different operating conditions, ships have different speeds and loads, and factors such as fuel consumption rate, combustion temperature and time will change, which in turn affects the emission of $NO_x$ from ships (Liu et al., 2022). $NO_x$ samples were collected under actual operating conditions of ships, and actual ship speed was monitored by the global positioning system (GPS) equipped on board. Samples were grouped based on three operating modes on the basis of the actual speed of each ship: cruising mode (> 8 knots, ship operating at higher speed), maneuvering mode (1–8 knots, ship operating at lower speed when approaching berths or anchorages) and hoteling mode (< 1 knot, ship at berth or anchored) (Chen et al., 2016). Meteorological conditions primarily affect ship emissions by influencing the operational mode of the ship, and this impact is relatively small and difficult to quantify (Huang et al., 2018; Duan et al., 2022; Zhao et al., 2020). Considering the unpredictable weather circumstances, we attempted to pick calm weather (with zephyrs and gentle waves) and appropriate temperature to carry out ship experiments. As presented in Table S1 of the Supporting Information (SI), the temperature, wind speed and relative humidity ranged from 1 to 27°C, from 2.8 to 5.1 m s$^{-1}$, and from 49 to 68 % during the observation period, respectively, which were obtained from the local weather station established by the China Meteorological Administration (Wang et al., 2019). Consequently, similar to some previous researches regarding $NO_x$ emitted from ships (Wang et al., 2019; Jiang et al., 2019; Zhang et al., 2018; Zhao et al., 2020), the influence of meteorological conditions during sampling was not taken into account in this study.

**Table 1.** Technical parameters of the test ships.

| vessel ID | engine power (kW) | rated speed (rpm) | maximum design speed (knot) | cylinders | gross tonnage (ton) | emission standard | ship length × width (m) | auxiliary engines | fuel |
|---|---|---|---|---|---|---|---|---|---|
| SH1 | 15748 | 75 | 14.5 | 6 | 94674 | Tier III | 292 × 45 | yes | residual oil and diesel |
| SH2 | 1470 | 850 | 11.52 | 6 | 6247 | Tier II | 109.8 × 26.8 | yes | residual oil and diesel |
| SH3 | 138.4 | 1150 | 8.45 | 4 | 77 | Tier I | 24 × 5.01 | no | diesel |
| SH4 | 120 | 1200 | 8 | 6 | 20 | Tier II | 28 × 4.8 | yes | diesel |
| SH5 | 178 | 1500 | 7 | 6 | 300 | Tier II | 35 × 6 | no | diesel |
| Y1 | 33 | 1500 | 7 | 4 | 5 | Tier I | 14 × 2.5 | no | diesel |
| Y2 | 29 | 1800 | 7 | 4 | 3 | before Tier I | 12 × 4 | no | diesel |
| K1 | 240 | 1900 | 20 | 6 | 30 | Tier II | 19.38 × 14.1 | no | diesel |
| KK1 | 610 | 750 | 11 | 6 | 499 | Tier II | 48.7 × 9 | yes | diesel |

A total of 146 $NO_x$ samples were collected in the present study. These $NO_x$ samples were collected directly from the chimney of ships to better reflect the initial condition of the emitted $NO_x$. $NO_x$ exhaust from the main engines (ME) of ships was collected. If the ship also has auxiliary engines (AE) and boilers, $NO_x$ emitted from AE were also collected, but boilers were not sampled since these make a small contribution to $NO_x$ emissions, less than 5% compared to those from ME and AE (Wan et al., 2020; Shi et al., 2020; Chen et al., 2017). The setup of the on-board sampling device was illustrated in Fig. S1 of SI. Before the ship emission test, a stainless steel bellow with a length of 1.5 or 3.0 m and an inner diameter of 40 mm was placed into the ship's chimney to direct the exhaust gas to the sampling platform. The exhaust gas was pumped at a flow rate of 1.0 L min$^{-1}$ into a gas-washing bottle containing 100 mL of 0.25 mol L$^{-1}$ potassium permanganate ($KMnO_4$) and 0.50 mol L$^{-1}$ sodium hydroxide (NaOH) absorption solution through a Teflon tube (approximately 1.5 m in length and 12.77 mm inner diameter), and $NO_x$ were collected as $NO_3^-$. Particulates and $HNO_3$ in the exhaust were removed when passing through a microporous filter and a Nylasorb filter, respectively before entering the gas-washing bottle. The method was proved to be effective at collecting 100 % ($\pm$5 %, 1$\sigma$) of the $NO_x$ and producing consistent isotope results under a wide variety of conditions (Fibiger et al., 2014; Zong et al., 2020a). Each sample was collected continuously for 20 min after 5 min of stable operation in each operating mode of the ship. An adequate sampling time is essential to ensure that sufficient amount of $NO_3^-$ was collected for conducting $\delta^{15}N$ measurement and minimizing uncertainties associated with isotope blank correction.

The whole sampling process was conducted carefully to avoid interference with isotopic measurements caused by $NH_x$, containing ammonia ($NH_3$) and ammonium ($NH_4^+$), and isotopic fractionation. The absorption solution prepared beforehand within 12 h before sampling was completely sealed before and after each collection, and titrated within 6 h after sampling to remove redundant $KMnO_4$ from the solution using 30 % hydrogen peroxide ($H_2O_2$) (Zong et al., 2020a). Fibiger and Margeson et al. found that the chemical conversion of $NH_3$ to $NO_3^-$ can lead to a 0.6 % or 2.8 % increase in $NO_3^-$ concentration when $KMnO_4$ was not removed from the absorption solution for 36 hours or 7 days, respectively (Fibiger et al., 2014; Margeson et al., 1984). The fast removal of $KMnO_4$ from the solution in the present study indicates that the experimental error regarding $NH_x$ was negligible. The total length of the connecting pipe (stainless steel bellows and Teflon tubes) from the ship chimneys to gas-washing bottles was about 2 m or 4 m for different ships and $NO_x$ were accordingly present in the pipe for less than 4 min, which was significantly shorter than the normal airborne NO lifespan ($\sim$15216.3 s), meaning that fractional distillation could be ignored (Massman, 1998). Penetration tests for $NO_x$ collection were performed on each vessel by connecting two gas-washing bottles in series with the same absorbent solution over a sampling period of 30 min. There was no experimental evidence of $NO_x$ penetration into the second bottle, denoting that the approach used in this study effectively collected all of the $NO_x$ from the sampled ship exhaust. Additionally, samples for background blank were collected during the sampling period of each ship (3 samples per ship, totaling 45 samples) to quantify background $NO_3^-$ concentrations and to correct for isotope blanks.

## 2.2 Chemical and isotopic analysis

The $NO_3^-$ concentration in the samples, where redundant $KMnO_4$ was removed, was quantified by standard colorimetric absorbance techniques (AutoAnalyzer 3, SEAL Analytical Ltd.) and the detection limit was 5 ng $mL^{-1}$. The bacterial denitrifier method was then conducted for $\delta^{15}N$–$NO_3^-$ analysis with the injection volume of samples calculated by the $NO_3^-$ concentration (Sigman et al., 2001; Mcilvin and Casciotti, 2011). In short, the collected $NO_3^-$ solution containing 20 nmol N was put into the 20 mL headspace bottle and then 2 mL concentrated bacterial solution (helium-purged at 30 mL $min^{-1}$ for 4 h to alleviate the background interference) was added to convert $NO_3^-$ to nitrous oxide ($N_2O$). *P. aureofaciens* was selected as the experimental strain, which lacks the $N_2O$ reductase enzyme. After sealed and reacting for 12 h, 0.1 mL 10 mol $L^{-1}$ NaOH was injected to terminate the denitrification process. Finally, the $\delta^{15}N$ of the produced $N_2O$ was analyzed by an isotope ratio mass spectrometer (MAT253, Thermo Fisher Scientific, Waltham, MA) and the $\delta^{15}N$ values were reported in parts per thousand relative to the international standards (IAEA–NO–3, USGS32, USGS34, and USGS35) (Bohlke et al., 2003):

$$\delta^{15}N = \left[ \frac{\left(^{15}N/^{14}N\right)_{sample}}{\left(^{15}N/^{14}N\right)_{standard}} - 1 \right] \times 1000 \tag{1}$$

The average $NO_3^-$ concentration of background blank samples from each ship was 45.33–682.50 ug N/L, accounting for $1.15 \pm 2.02$ % of that for the regular samples. This includes the background value of $NO_3^-$ in the absorption solution, and the $NO_3^-$ converted from $NO_x$ captured from ambient air during the collection time of 20 minutes. Therefore, the final $NO_3^-$ concentrations for samples from each vessel were recalculated by subtracting the average blank value during sampling. The average $\delta^{15}N$ values related to the background blank for each ship ranged from -3.02 ‰ to 11.34 ‰ and $\delta^{15}N$ value for each sample was redetermined by mass balance (Fibiger et al., 2014), leading to an average variation in $\delta^{15}N$ values ranging from 1.12 % to 4.87 %:

$$\delta^{15}N = \frac{\delta^{15}N_{total}[NO_3^-]_{total} - \delta^{15}N_{blank}[NO_3^-]_{blank}}{[NO_3^-]_{total} - [NO_3^-]_{blank}} \tag{2}$$

where $\delta^{15}N_{total}$ and $\delta^{15}N_{blank}$ are the $\delta^{15}N$ values (%) of samples and blanks of the ship, respectively; $[NO_3^-]_{total}$ and $[NO_3^-]_{blank}$ are $NO_3^-$ concentrations (ug N/L) of samples and blanks of the ship, respectively. The analytical precision of $NO_3^-$ concentrations and $\delta^{15}N$ values were less than 1.8 % and 0.5 ‰, respectively, as determined by the replicates in this study.

## 2.3 Data analysis

Considering the high power and load, ME of ships are the main $NO_x$ source, and emissions vary greatly between operating conditions. AE drive other power machinery on board besides ME, such as generators, oil splitters, marine pumps and air conditioning units, and the output power usually varies with the ME power during navigation. Hence the AE to ME Power Ratio (typically 0.22) was used to estimate the rated power of AE (Chen et al., 2017; Trozzi, 2010). Since a ship's ME and AE work simultaneously for most times, the emission powers of the two were used for weighted calculations, and thus the actual $\delta^{15}N$–$NO_x$ values emitted by ships under each operating condition could be obtained as follows:

$$\delta^{15}N = \frac{0.22 \times \delta^{15}N_{AE} + LF \times \delta^{15}N_{ME}}{0.22 + LF} \tag{3}$$

where $\delta^{15}N_{AE}$ and $\delta^{15}N_{ME}$ are the $\delta^{15}N-NO_x$ values (‰) emitted by AE and ME of the ship, respectively; LF (%) is the load factor of ME under different operating conditions of the ship and can be calculated from the actual sailing speed (AS, knot) and the maximum design speed (MS, knot) of the ship (Chen et al., 2017):

$$LF=\left(\frac{AS}{MS}\right)^3 \tag{4}$$

It was capped to 1.0 in the case the calculated value exceeded 100%. Details on $NO_x$ emissions of collected ship exhaust after integration of ME and AE are shown in Table S2 of SI.

Among the statistical analysis methods, the analysis of variance (ANOVA) and the Mann–Whitney U test were applied to support the factors influencing $\delta^{15}N-NO_x$ values from ships and further compare whether there was significant discrepancy between different classifications under the influence factor, respectively. ANOVA is a statistical method used to compare the means of two or more groups to determine if there is a significant difference. The core idea of ANOVA is to compare the differences between groups to the differences within groups. It calculates the ratio of between-group variation to within-group variation, known as the $F$ value, and then compare this $F$ value with the $F$ critical value corresponding to the given significance level to determine if the mean differences are statistically significant. The Mann–Whitney U test is widely used in various fields for evaluating differences between two groups by comparing whether the medians of two samples are the same (Mann and Whitney, 1947). The Mann-Whitney U test typically reports a $p$ value, which represents the probability of observing the current test statistic or a more extreme test statistic under the same null hypothesis conditions. If the $p$ value is less than the significance level, the null hypothesis that the medians of the two samples are the same can be rejected.

The conditional inference tree (CIT), random forest (RF), and boosted regression tree (BRT) were implemented to quantitatively evaluate the impact degree of different factors on the variation in ship-emitted $\delta^{15}N-NO_x$ values. 75 % of the sample data were utilized to generate the prediction model, and the remaining data were utilized to evaluate the accuracy of the simulation results of the prediction model. The CIT is a non-parametric decision tree algorithm that recursively binary splits the dependent variable based on the values of correlations. It can handle features with different scales and selects features in an unbiased manner, as the feature and the best split point are determined after the feature selection (Hothorn et al., 2006). RF is an ensemble of regression trees originally used for classification. It evaluates the importance of candidate predictor variables by measuring the variance reduction in predictive accuracy before and after permuting the variables and combines the predictions of multiple trees to improve the overall model's performance (Strobl et al., 2007; Speybroeck, 2012). The BRT combines the strengths of regression trees and boosting, which is an adaptive method that combines many simple models to improve predictive performance (Elith et al., 2008). In the operation process, BRT randomly selects a subset of data multiple times to analyze the impact of predictor variables on the dependent variable and uses the remaining data to validate the fitting results. The output is the average of the generated regression trees. BRT is tolerant to covariance among predictors and non-normality, and it is less prone to overfitting, thus providing higher predictive accuracy for new data. These statistical analysis methods were conducted by R 4.1.3 software.

## 3 Results and discussion

### 3.1 $\delta^{15}N–NO_x$ values emitted from ships

As illustrated in Fig. 1, the $\delta^{15}N–NO_x$ values emitted from ships sampled in this study were in the range of −35.8 ‰ to 2.04 ‰, with a mean ± standard deviation of −18.5 ± 10.9 ‰. There can be large variation in $\delta^{15}N–NO_x$ values emitted from the same vessel under the same operating condition as shown in Table S2, which was often observed during hoteling and cruising mode. This could be attributed to changes in the usage requirements of different onboard equipment in hoteling mode, as well as the differences in engine load concerning the substantial variations in vessel speed during cruising mode (Cooper, 2003; Huang et al., 2018). A more detailed interpretation can be found in Text S1 of SI. $NO_x$ produced during the combustion of fossil fuels can be divided into two groups: fuel $NO_x$, generated when chemically bound nitrogen in fuel oxidizes, and thermal $NO_x$, which is related to the thermal immobilization of atmospheric nitrogen ($N_2$), i.e., produced by the reaction between oxygen and nitrogen in air at high temperatures (Beyn et al., 2015; Walters et al., 2015b). Previous measurements have revealed that $NO_x$ derived from biomass burning and coal combustion are inclined to be enriched in $^{15}N$ abundance, largely influenced by the nitrogen content of the biomass and coal itself (Felix et al., 2012; Fibiger and Hastings, 2016; Zong et al., 2022), and $NO_x$ produced thermally by internal combustion engines tends to be depleted in $^{15}N$ abundance due to the kinetic isotope effect associated with the thermal decomposition of the strong triple bond of $N_2$ (Ti et al., 2021; Walters et al., 2015a; Walters et al., 2015b; Zong et al., 2020a; Snape et al., 2003). Moreover, it is well-known that residual oil as marine fuel remains after the removal of valuable distillates (such as gasoline) from petroleum, which contains more impurities, including nitrogen containing substrates, than gasoline and diesel (Corbett and Winebrake, 2008). If the nitrogen-containing substances in the residual oil are the major contributors to the vessel-emitted $NO_x$, it is expected that these $NO_x$ may be enriched in $^{15}N$ abundance (Felix et al., 2012; Fibiger and Hastings, 2016). However, we found that there was no statistically significant difference in $\delta^{15}N–NO_x$ values emitted from vessels fueled by residual oil and diesel at the 99 % confidence level ($p > 0.01$), although the former (−14.7 ± 7.72 ‰, n = 14) was slightly higher than the latter (−18.9 ± 11.1 ‰, n = 109) as indicated in Fig. S2 of SI. Therefore, it can be concluded that the $\delta^{15}N–NO_x$ values emitted from ships do not depend much on the fuel type, and this finding is in agreement with the small proportion (10 %) of $NO_x$ originating from fuel-combined nitrogen for engines burning residual oil reported in a previous study (Goldsworthy, 2003). Overall, the negative $\delta^{15}N–NO_x$ values emitted from ships in this research, with minimal influence from fuels, suggest that these values are more likely associated with the production of thermal $NO_x$ rather than the conversion of nitrogen in the fuels (Walters et al., 2015a; Walters et al., 2015b; Zong et al., 2022).

It was found that the majority of the $NO_x$ emissions from cars are derived from thermal production (Toof, 1986; Tsague et al., 2006). Figure 1 summarizes the $\delta^{15}N–NO_x$ values emitted from ships sampled in this study and from vehicles reported in previous studies to assess whether there are significant differences in $\delta^{15}N$ values of these thermally generated $NO_x$ (Walters et al., 2015a; Walters et al., 2015b). Statistical results showed that the $\delta^{15}N–NO_x$ values were −12.3 ± 7.21 ‰ (n = 51), −6.13 ± 6.59 ‰ (n = 158) and −0.100 ± 1.76 ‰ (n =3) for diesel-, gasoline- and liquefied petroleum gas (LPG)-powered

combustion engines, respectively. The $\delta^{15}N–NO_x$ values from ships ($-18.5 \pm 10.9$ ‰, n = 123) were

significantly lower than those produced by diesel-, gasoline- and LPG-powered combustion engines of

vehicles at a confidence level of 99 % ($p < 0.01$), as shown in Fig. 1. The significant difference observed

highlights the importance of addressing the data gap in measuring $\delta^{15}N–NO_x$ emissions from ships for

accurately apportioning atmospheric $NO_x$ sources in coastal zones, especially in some port areas with

high ship activity. Further discussion is needed to understand the reasons behind the noticeable difference

and to better utilize the measured values of ship-emitted $\delta^{15}N-NO_x$, including the variations in $\delta^{15}N–NO_x$

values and their primary influencing factors.

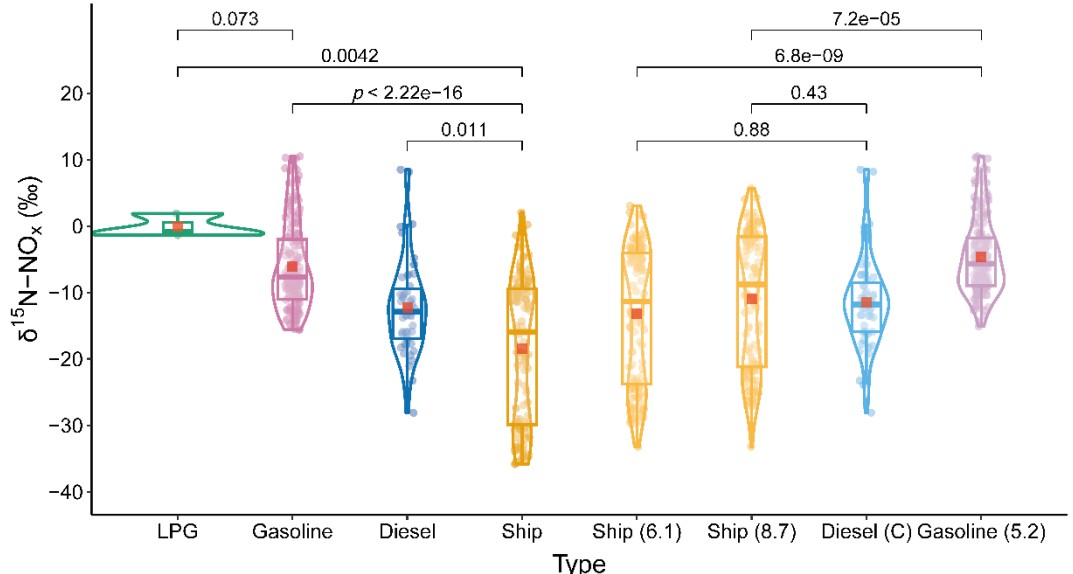

**Figure 1.** $\delta^{15}N–NO_x$ values emitted from ships in this study and cars fueled by diesel, gasoline and liquefied
petroleum gas (LPG) reported from other references (Walters et al., 2015a; Walters et al., 2015b). The Ship (6.1) and
Ship (8.7) indicate that $\delta^{15}N–NO_x$ values from ships without selective catalytic reduction (SCR) systems are
corrected by 6.1 ‰ and 8.7 ‰, respectively. The Diesel (C) and Gasoline (5.2) indicates that $\delta^{15}N–NO_x$ values from
diesel and gasoline cars without SCR systems are corrected using 6.1 ‰ or 8.7 ‰, and 5.2 ‰, respectively. (red
square, mean; center line, median; box bounds, upper and lower quartiles; whiskers, 1.5 times interquartile range;
points, outliers; outer line, data distribution). The $p$ values indicating the distinction between two groups are marked
on the upper of the panel (the Mann–Whitney U test).

### 3.2 Main factors affecting $\delta^{15}N–NO_x$ emitted by ships

The $\delta^{15}N–NO_x$ values emitted from ships had a wider range of variation than those from other sources in

the source apportionment of $NO_x$, which is not conducive to constraining the sources of $NO_x$ in the

atmosphere (see Table S3 of SI) (Snape et al., 2003; Felix and Elliott, 2014; Li and Wang, 2008; Moore,

1977; Chai et al., 2019; Redling et al., 2013; Zong et al., 2022; Felix et al., 2012; Fibiger and Hastings,

2016; Walters et al., 2015a; Walters et al., 2015b; Yu and Elliott, 2017; Felix and Elliott, 2013; Miller et

al., 2018; Miller et al., 2017; Perez et al., 2001; Ammann et al., 1999; Shi et al., 2022; Freyer, 1978;

Heaton, 1990). Clarifying the main factors affecting the change of $\delta^{15}N–NO_x$ is beneficial to narrow the

range of $\delta^{15}N–NO_x$ changes in source analysis. Several classification indicators including the emission

regulation met by ship engines, the ship category, the ship fuel type, and the actual operational status of

ships are considered for the assessment in this study. The reason for considering compliance with the

emission regulation for ship engines as a criterion is that the IMO emission regulation is the most

important measure to restrict $NO_x$ emissions from ships and once each standard was proposed by the

IMO, newly constructed or significantly refurbished marine diesel engines must comply with its requirements. Therefore, the emission regulation that a ship complies with is actually related to its age. Besides, the category, fuel type, and actual operational status are often used to assess the variation in $\delta^{15}$N–NO$_x$ values from vehicles (Walters et al., 2015a; Walters et al., 2015b). The statistics of $\delta^{15}$N–NO$_x$ values classified according to the four indicators are illustrated in Fig. S2–S4 of SI and Fig. 2. Various analytical methods were adopted to explore the impact of the four indicators on ship emission $\delta^{15}$N–NO$_x$ values. The outcomes of ANOVA are shown in Table 2. The $p$ values of $\delta^{15}$N–NO$_x$ values grouped by the emission regulation and vessel category from the variance analysis were both less than 0.001, indicating that the two indicators were the dominant factors influencing the variation in ship $\delta^{15}$N–NO$_x$ values. Similar significant differences (small $p$ values) in $\delta^{15}$N–NO$_x$ values between types of emission regulations, and different vessel categories were calculated by the Mann–Whitney U test, as displayed in Fig. 2 and Fig. S3, respectively.

**Table 2.** Results of ANOVA.

| classification indicators | degree of freedom | sum of squares | mean of squares | $F$ | $p$ |
|---|---|---|---|---|---|
| vessel category | 3 | 3644 | 1214.5 | 19.343 | 3.30E−10 |
| emission regulation | 2 | 3070 | 1534.8 | 24.444 | 1.46E−09 |
| fuel type | 1 | 138 | 138.4 | 2.205 | 0.1403 |
| operational status | 2 | 380 | 189.9 | 3.025 | 0.0525 |

The CIT analysis provided a more intuitive result as indicated in Fig. 3. It was found that the emission regulation met by ships was the most important splitting factor of the root node (node 1) and the second terminal node. Samples collected from ships prior to the implementation of IMO Tier I (stage 1) and ships implementing Tier I–III (stage 2–4) were separated to terminal node 9 (−33.8 ± 1.83 ‰) and node 2 (−16.8 ± 9.50 ‰), respectively, indicating a strong impact of implementing Annex V to the MARPOL 73/78 on these $\delta^{15}$N–NO$_x$ values. Then the emission regulation was again used as the next splitting factor to separate $\delta^{15}$N–NO$_x$ values emitted from ships meeting Tier III (stage 4) from node 2 as node 3 (−7.93 ± 5.33 ‰). The $\delta^{15}$N–NO$_x$ values emitted from ships meeting Tier I & II (stage 2 and 3) as node 4 (−18.3 ± 9.25 ‰) were subsequently divided by the ship category to node 5 (passenger ship and research ship, −11.3 ± 1.67 ‰) and node 6 (cargo ship and fishing boat, −21.0 ± 9.48 ‰). Finally, fuel type was the last splitting variable, separating samples taken when the ships used diesel (node 7, −22.7 ± 9.25 ‰) and residual oil (node 8, −11.2 ± 1.38 ‰) as fuel. The splitting process suggests that the emission regulation has a greater influence on $\delta^{15}$N–NO$_x$ values than the ship category. Analogously, the results of the RF and BRT methods elucidated that the most important influencing factor on $\delta^{15}$N–NO$_x$ values from ships was emission regulation, followed by ship category, fuel type, and operational status, as shown in Fig. S5 and Fig. S6 of SI. The relevant parameters for the accuracy assessment of these three decision tree–based methods are listed in Table S4 of SI. The influence of ship category on ship-emitted $\delta^{15}$N–NO$_x$ values primarily concerns engine types of different ships and the minor influence of fuel type is due to the principle of thermally generated NO$_x$ by internal combustion engines of ships as mentioned above (Goldsworthy, 2003). The operational condition of ships has the least effect on the variation in $\delta^{15}$N–NO$_x$ values. More detailed interpretations can be found in Text S2 of SI.

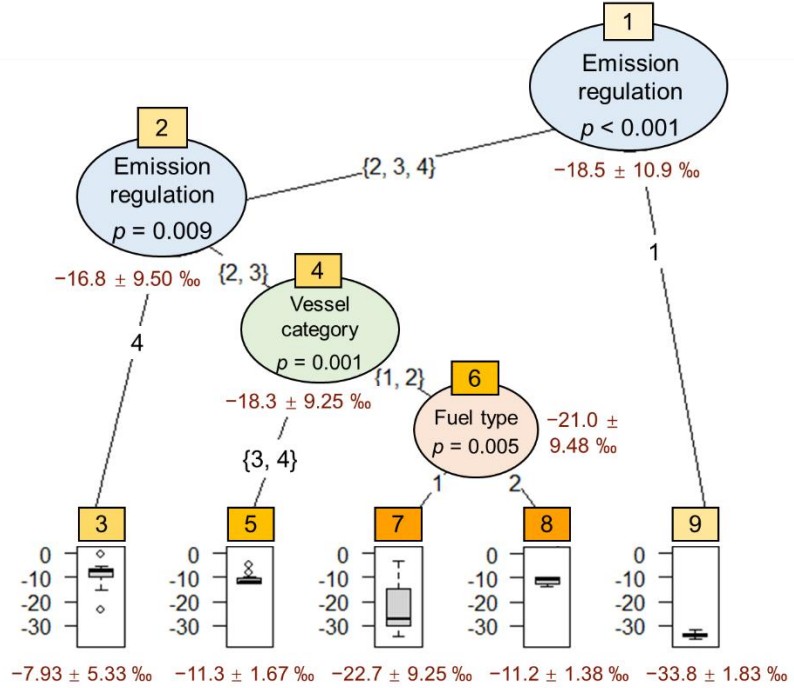

**Figure 3.** Conditional inference trees (CIT) for the $\delta^{15}$N–NO$_x$ values emitted from ships. For each inner node, the $p$ values are given and the range of $\delta^{15}$N–NO$_x$ values is displayed for each terminal node.

Figure 2 displays $\delta^{15}$N values in NO$_x$ emitted from ships under different emission regulations established by the IMO and more information on ship-emitted NO$_x$ at various stages of emission regulations is summarized in Table 3. The mean ± standard deviation of $\delta^{15}$N–NO$_x$ values and their 95 % confidence intervals estimated using bootstrapping were −33.8 ± 1.83 ‰ (n=12, −34.8 – −32.7 ‰), −21.5 ± 6.67 ‰ (n=12, −25.2 – −17.7 ‰), −17.8 ± 9.88 ‰ (n=83, −19.9 – −15.4 ‰), and −8.12 ± 8.84 ‰ (n=16, −10.8 – −5.64 ‰) for the stage 1, 2, 3 and 4, respectively. In general, the $\delta^{15}$N–NO$_x$ values progressively increased with the tightening of ship NO$_x$ emission standards. This is in accordance with the variation trend of $\delta^{15}$N–NO$_x$ values emitted from gasoline vehicles complying with the national vehicle emission standards GB I to GB VI in China, which were established by the Ministry of Ecology and Environment of the People's Republic of China (Zong et al., 2020a). The largest average growth rate of $\delta^{15}$N–NO$_x$ values emitted from ships between the three adjacent phases from 1 to 4 occurred between the implementation of IMO Tier II and III (stage 3–4, 54.1 %), followed by before and after the implementation of Tier I (stage 1–2, 36.4 %), and between Tier I and II (stage 2–3, 17.8 %). Since the implementation of IMO Tier I on January 1, 2000, fuel optimization technologies (fuel emulsification, fuel desulfurization and fuel additives) and pre-combustion control technologies (fuel injection strategy, water injection strategy, Miller cycle, two-stage turbocharging, and dual-fuel combustion strategy), as demonstrated in Table 3, have been applied and experienced rapid development. The former reduce the production of NO$_x$ by changing the composition of fuel or adding additives to improve the combustion process, while the latter primarily suppress the formation of NO$_x$ by modifying the internal structure of diesel engines or adjusting parameters (Ampah et al., 2021; Deng et al., 2021; Lion et al., 2020). The average growth rate of $\delta^{15}$N–NO$_x$ values between Tier I and Tier II (17.8 %) is lower than that before and after the implementation of Tier I (36.4 %). This difference can be attributed to the fact that different emission standards have varying levels of impact on NO$_x$ emission reductions. Compared to the

optimization and development of emission reduction technologies between Tier I and II, the implementation of Tier I fills the gap in ship engine and fuel emission reduction measures, which clearly has a greater impact on $NO_x$ emissions. The exhaust after-treatment system, including exhaust gas recirculation (EGR), selective catalytic reduction (SCR), non-thermal plasma (NTP), etc., has been widely adopted after the release of IMO Tier III. It offers the best emission reduction results by effectively reducing $NO_x$ emissions from marine diesel engines while maintaining engine performance and fuel efficiency. As the most commonly used exhaust after-treatment system, SCR can reduce $NO_x$ emissions by more than 90 % without increasing fuel consumption by injecting a urea solution into the exhaust gas and using a catalyst to convert the $NO_x$ in the exhaust gas into harmless nitrogen and water. Previous studies have found significant differences in $\delta^{15}N–NO_x$ emissions from fuel combustion sources with and without SCR systems because lighter molecules of $NO_x$ are preferentially decomposed during catalytic reduction, so the produced $NO_x$ are inclined to be enriched in $^{15}N$ abundance because of balanced isotope effects (Felix et al., 2012). Comparably, distinctions were also observed in $\delta^{15}N–NO_x$ values released by vehicles equipped with three-way catalytic (TWC) converters, which made the $\delta^{15}N–NO_x$ values from those vehicles reliant on operating circumstances (e.g., cold start or warm start) and $NO_x$ mitigation efficiency (Walters et al., 2015a; Walters et al., 2015b).Consequently, the progressive adoption of $NO_x$ emission control technologies (e.g., SCR and TWC) is anticipated to bring about a rise in the $\delta^{15}N–NO_x$ values of $NO_x$ emissions associated with fuel combustion, and the magnitude of this increase depends on the $NO_x$ fractionation characteristics and the efficiency of $NO_x$ reduction associated with the catalytic reduction technology employed (Felix et al., 2012). Accordingly, the $\delta^{15}N–NO_x$ values from ships equipped with SCR systems, which comply with the Tier III emission standard, were significantly higher than those from ships equipped without SCR systems at the 99 % confidence level ($p < 0.01$) in our study; therefore, the relatively largest difference in $\delta^{15}N–NO_x$ values between ships implementing Tier II and Tier III, as shown in Fig. 2, is in a great measure derived from the SCR system.

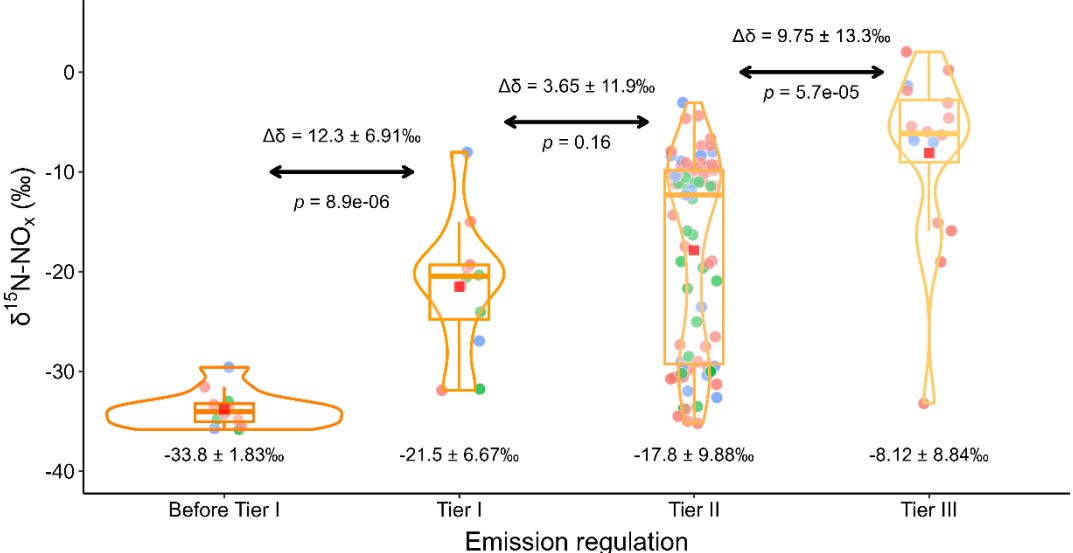

**Figure 2.** $\delta^{15}N–NO_x$ values emitted from ships under different emission regulations established by the IMO. (red square, mean; center line, median; box limits, upper and lower quartiles; whiskers, 1.5 times interquartile range; points, outliers (pink, cruising; green, hoteling; blue, maneuvering); outer line, distribution of data). Mean ± standard

deviation of $\delta^{15}N$–$NO_x$ values of each group is marked on the bottom of the panel. The difference and $p$ values indicating the distinction between two groups are marked on the upper of the panel (the Mann–Whitney U test).

**Table 3.** Summary of ship-emitted $NO_x$ at each stage of emission regulations.

| stage | effective date of the emission regulation | $NO_x$ emission limits (g $(KWh)^{-1}$) | | | $NO_x$ emission reduction technologies | $NO_x$ (ppm) | $\delta^{15}N$–$NO_x$ (‰) |
|---|---|---|---|---|---|---|---|
| | | $N^a <$ 130 | $130 \leqslant N$ $< 2000$ | $N \geqslant$ 2000 | | | |
| Before Tier I | | | | | | 96.6 ± 28.6 | −33.8 ± 1.83 |
| Tier I | 2000/01/01 | 17.0 | $45N^{-0.2}$ | 9.8 | fuel optimization: fuel emulsification, fuel desulfurization, fuel additives; pre-combustion control technologies: fuel injection, water injection, Miller cycle, two-stage turbocharging, dual-fuel combustion | 239 ± 93.0 | −21.5 ± 6.67 |
| Tier II | 2011/01/01 | 14.4 | $44N^{-0.23}$ | 7.7 | | 169 ± 156 | −17.8 ± 9.88 |
| Tier III (in NECAs[b]) | 2016/01/01 | 3.4 | $9N^{-0.2}$ | 1.96 | exhaust after-treatment system: exhaust gas recirculation (EGR), selective catalytic reduction (SCR), non-thermal plasma (NTP), seawater flue gas desulphurization (SWFGD) | 122 ± 91.6 | −8.12 ± 8.84 |

[a] N: rated speed (rpm)

[b] NECAs: the Nitrogen Oxide Emission Control Areas, mainly including the Baltic Sea area, the North American Emission Control Area, and the Caribbean Sea Emission Control Area.

Meanwhile, compared to the $\delta^{15}N$–$NO_x$ values which progressively increased with the tighter policing, the corresponding $NO_x$ concentrations gradually decreased and were 239 ± 93.0 ppm, 169 ± 156 ppm, and 122 ± 91.6 ppm under Tier I, II and III, respectively, as presented in Table 3. These gradually increasing $\delta^{15}N$–$NO_x$ values and decreasing $NO_x$ emissions showed a negative logarithmic correlation

($r = -0.39$, $p < 0.01$), as shown in Fig. 4. Walters et al. (2015b) demonstrated that there was also a negative logarithmic relationship between $\delta^{15}N$–$NO_x$ values and $NO_x$ concentrations from vehicle emissions, which was stronger in vehicles with $NO_x$ emission control technologies ($r = -0.92$) than in vehicles without $NO_x$ emission control technologies ($r = -0.1$). The stronger relationship between $\delta^{15}N$–$NO_x$ values and $NO_x$ concentrations for vehicles with $NO_x$ emission control technologies versus vehicles

without, is attributed to the enrichment of $\delta^{15}N$ relative to the thermally produced $NO_x$ while catalytically reducing $NO_x$ to $N_2$ (Walters et al., 2015a). The strength of the correlation between $\delta^{15}N$–$NO_x$ values and $NO_x$ concentrations of ship emissions is just within the range of those reported in vehicle emissions with or without $NO_x$ emission control technologies because only 13 % of ship-emitted samples in our research were collected from ships equipped with SCR systems, further revealing that whether a ship is equipped

with the $NO_x$ catalytic reduction device is a major factor that affects the $\delta^{15}N$–$NO_x$ values.

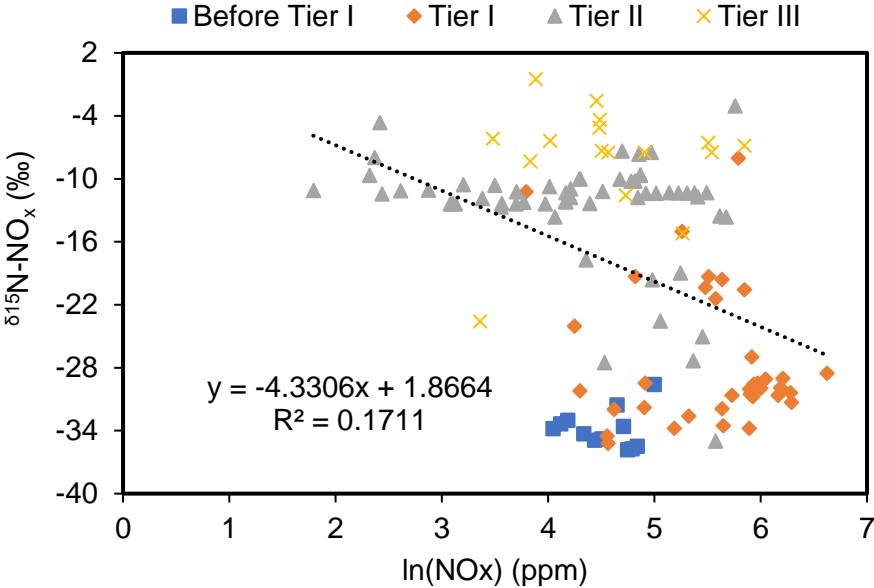

**Figure 4.** The negative logarithmic relationship between $\delta^{15}N–NO_x$ values and $NO_x$ concentration emitted from ships.

To further determine the effect of exhaust gas treatment devices on the emission of thermal $\delta^{15}N–NO_x$ values from different internal combustion engines, the $\delta^{15}N–NO_x$ values from ships, diesel vehicles and gasoline vehicles not equipped with $NO_x$ catalytic reduction devices were corrected separately and compared again as shown in Fig. 1. It was found that the enrichment factors concerning catalytic $NO_x$ reduction relative to the original thermal $NO_x$ were 6.1 ‰ and 8.7 ‰ for light- and heavy-duty diesel-powered engines and 5.2 ‰ for gasoline-powered engines (Walters et al., 2015a; Walters et al., 2015b). Consequently, the $\delta^{15}N–NO_x$ values emitted from ships without catalytic reduction systems were corrected by the enrichment factors of 6.1 ‰ and 8.7 ‰, and diesel vehicles and gasoline vehicles without catalytic reduction systems were corrected by 6.1 ‰ or 8.7 ‰, and 5.2 ‰, respectively. After the correction, the $\delta^{15}N–NO_x$ values from ships are $-11.0 \pm 10.1$ ‰ (corrected by 8.7 ‰) and $-13.2 \pm 10.3$ ‰ (corrected by 6.1 ‰), and those from diesel vehicles and gasoline vehicles are $-11.5 \pm 6.87$ ‰ and $-4.62 \pm 5.71$ ‰, respectively. These corrected $\delta^{15}N–NO_x$ values from ships are still significantly lower than the $\delta^{15}N–NO_x$ values from gasoline vehicles but insignificantly different than those from diesel vehicles ($p > 0.1$), demonstrating that the catalytic reduction system is a major reason for the differences in $\delta^{15}N–NO_x$ values produced by various internal combustion engines. In addition, we found no significant difference in the $\delta^{15}N–NO_x$ values emitted from gasoline and LPG vehicles. Hence the $\delta^{15}N–NO_x$ values after eliminating the effect of SCR systems can be divided into two groups, one from ships and diesel vehicles and the other from gasoline and LPG vehicles. The $\delta^{15}N–NO_x$ values in the first group are emitted from compression ignition engines fueled by diesel and residual oil, and those in the second group are emitted from spark-ignition engines fueled by gasoline and LPG (Mikalsen, 2011; Mitukiewicz et al., 2015; Park et al., 2020). Engines of the same structural design using different fuels produced comparable $\delta^{15}N–NO_x$ values, which indicates that fuel type is not a major factor affecting the $\delta^{15}N–NO_x$ values once more (Mikalsen, 2011). The main reasons for the significant difference in $\delta^{15}N–NO_x$ values between the two groups may be the difference in the state variables during operation of the two different engine designs.

It has been well proven that the combustion with high pressure, the extended Zeldovich mechanism and $N_2O$ reactions are the major sources of combustion engine-emitted $NO_x$ (Goldsworthy, 2003). The combustion chamber temperature, equivalence ratio of fuel mass to oxidizer mass, oxygen concentration and retention time of oxygen and nitrogen at high temperature are different between the types of engines and are the main factors determining the $NO_x$ generation rate (Cho et al., 2018; Goldsworthy, 2003; Mikalsen, 2011; Querel et al., 2015; Tomeczek and Gradon, 1997). These parameters are likely to be the major causes of the large variations in $\delta^{15}N–NO_x$ values, and should be further quantified in future studies.

### 3.3 Implications for $\delta^{15}N–NO_x$ values from ships

As mentioned in Text S3 of SI, the impact of ship emissions on atmospheric $NO_x$ pollution cannot be ignored (Fig. S7 of SI), and reliable $\delta^{15}N–NO_x$ values of ship emissions are essential for the accuracy of source apportionment when assessing atmospheric $NO_x$ sources in coastal areas based on $\delta^{15}N$ methods. Although the $\delta^{15}N–NO_x$ values obtained in the present study are constrained and do not accurately reflect the data emitted from all types of ships, they are practical for assessing $\delta^{15}N–NO_x$ source characterization of traffic exhaust. In general, we found that ships and diesel vehicles emit lower $\delta^{15}N–NO_x$ values than gasoline and LPG vehicles because these $NO_x$ are produced by compression ignition engines, and $\delta^{15}N–NO_x$ values emitted from ships increase with the decrease in the emission factor of $NO_x$ to meet the requirements of tightened regulations established by the IMO, especially when ships are equipped with SCR systems. A previous study concluded that the volume of global maritime trade increased by almost 3-fold from 1980 to 2019, and approximately 90 % of these goods were carried by merchant fleets (Kong et al., 2022). Thus, this study collected the age distribution of ships larger than 300 gross tonnage (GT) in the international merchant fleet during 2001 and 2021 to assess the temporal variation in $\delta^{15}N–NO_x$ emitted from ships by developing a mass-weighted model (Yang and Zhou, 2002; Zhou et al., 2004; Meng et al., 2005; Meng and Huang, 2006; Meng et al., 2007; Qi et al., 2008; Qin et al., 2009; Qi et al., 2010; Li et al., 2011; Qin et al., 2012; Qi et al., 2013; Li et al., 2014; Qin et al., 2015; Li et al., 2016; Qin et al., 2017, 2018; Shen and Qi, 2019; Qi et al., 2020; Liu et al., 2021; Wei et al., 2022). In the model, the mass-weighted $\delta^{15}N–NO_x$ value in a specified year can be expressed as:

$$\delta^{15}N = \frac{\sum_{i=1}^{4}\left(\delta^{15}N_i \times EF_i \times TS_i\right)}{\sum_{i=1}^{4}(EF_i \times TS_i)} \tag{5}$$

where $\delta^{15}N_i$, $EF_i$, and $TS_i$ are the $\delta^{15}N–NO_x$ value, emission factor of $NO_x$, and number of ships complying with the $i^{th}$ emission regulation, respectively. The subscript of i from 1 to 4 means before Tier I, Tier I, Tier II, and Tier III, respectively. The $\delta^{15}N–NO_x$ values in the four stages are shown in Fig. 2. The EF values in the four stages are 9.8 g $(KWh)^{-1}$, 9.8 g $(KWh)^{-1}$, 7.7 g $(KWh)^{-1}$, and 1.96 g $(KWh)^{-1}$, as presented in Table 3 (https://dieselnet.com/standards/inter/imo.php#other). The TS was calculated based on the age distribution of ships during 2001 and 2021, as shown in Fig. S8 of SI. On this basis, 100000 $\delta^{15}N–NO_x$ values were stochastically generated to be the typical values of ships meeting each emission regulation to calculate the mass-weighted $\delta^{15}N–NO_x$ value in a specified year.

The temporal variation in the mass-weighted $\delta^{15}N–NO_x$ emitted from ships between 2001 and 2021 is displayed in Fig. 5, and the specific calculation results of the mass-weighted $\delta^{15}N–NO_x$ values used in this figure can be found in Table S5 of SI. As expected, the $\delta^{15}N–NO_x$ values from ships larger than 300

GT in the international merchant fleet continued to increase as ships implemented tightener emission standards. The growth rate of the $\delta^{15}$N–NO$_x$ values was relatively gentle during approximately ten years from 2001 to 2012 (0.25 ‰/yr), and then became faster between 2013 and 2021 (0.40 ‰/yr) attributed to the further tightening of emission regulations. Assuming the same age distribution of vessels after 2022 as in 2021, it can be estimated that the $\delta^{15}$N–NO$_x$ values will continue to increase before 2040

(0.40 ‰/yr) as shown in Fig. 5. After a few years of gentle growth, the $\delta^{15}$N–NO$_x$ values will remain basically flat after 2046 when all ships will meet the Tier III emission standard. In conclusion, the $\delta^{15}$N–NO$_x$ values obtained after mass weighting based on the ship age distribution have a smaller range of variation and more valuable and practical significance. In addition, given that the calculated results only involved the age distribution and emission reduction level of the international merchant fleet, the

subsequent process of using $\delta^{15}$N to evaluate the contribution of ship emissions to atmospheric NO$_x$ can be combined with the actual situation of ships in the study area to select more appropriate $\delta^{15}$N–NO$_x$ values to acquire a more accurate ship emission contribution and reduce the uncertainty in NO$_x$ source apportionment.

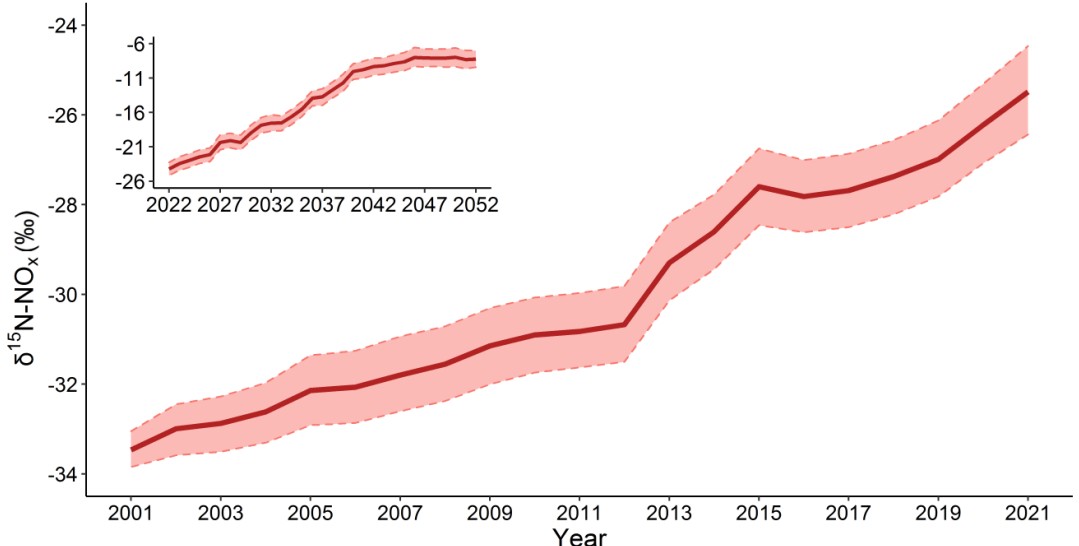

**Figure 5.** Temporal variation in the mass-weighted $\delta^{15}$N values in NO$_x$ emitted from ships larger than 300 GT in the international merchant fleet during 2001 and 2021. Predicted temporal changes in $\delta^{15}$N–NO$_x$ values from these ships between 2022 and 2052 assuming that the distribution of the ship ages is the same as that in 2021 are given in the inset. (red solid line, mean; red dashed line, upper and lower quartiles)

## 4 Summary and conclusions

The $\delta^{15}$N–NO$_x$ values emitted from ships play a crucial role in accurately analyzing the sources of atmospheric NO$_x$ in coastal areas with frequent shipping activities, as well as the sources of nitrogen deposition in remote marine regions. This study conducted the first measurements of $\delta^{15}$N–NO$_x$ values emitted from ships and investigated their main influencing factors. Moreover, a weighted model that enabled precise assessment of the temporal variation in ship-emitted $\delta^{15}$N–NO$_x$ values was developed

based on the relationship between observed $\delta^{15}$N–NO$_x$ values and emission regulations. Using this model, the $\delta^{15}$N–NO$_x$ values emitted from ships in the international merchant fleet was calculated with the age

distribution of ships in the fleet.

The $\delta^{15}$N–NO$_x$ values emitted from ships sampled in this study were $-18.5 \pm 10.9$ ‰, significantly lower ($p < 0.01$) than the reported $\delta^{15}$N–NO$_x$ values from vehicle emissions. Given that these values were negative and there was no significant statistical difference ($p > 0.01$) between $\delta^{15}$N–NO$_x$ values emitted from ships using residual oil and diesel fuel, the $\delta^{15}$N–NO$_x$ values are more likely associated with the production of thermal NO$_x$ rather than the conversion of nitrogen in the fuels.

Statistical analysis methods and machine learning techniques indicated that the most important influencing factor on $\delta^{15}$N–NO$_x$ values from ships was emission regulation, followed by ship category, fuel type, and operational status. Overall, the $\delta^{15}$N–NO$_x$ values progressively increased with the tightening of ship NO$_x$ emission standards. The $\delta^{15}$N–NO$_x$ values from ships equipped with SCR systems, which comply with the Tier III emission standard, were significantly higher than those from ships equipped without SCR systems at the 99 % confidence level ($p < 0.01$). Therefore, the relatively largest difference in $\delta^{15}$N–NO$_x$ values between ships implementing Tier II and Tier III is in a great measure derived from the SCR system. Furthermore, the $\delta^{15}$N–NO$_x$ values from ships, diesel vehicles and gasoline vehicles not equipped with NO$_x$ catalytic reduction devices were corrected separately and compared again. These corrected $\delta^{15}$N–NO$_x$ values from ships are still significantly lower than the $\delta^{15}$N–NO$_x$ values from gasoline vehicles but insignificantly different than those from diesel vehicles ($p > 0.1$), demonstrating that the catalytic reduction system is a major reason for the differences in $\delta^{15}$N–NO$_x$ values produced by various internal combustion engines.

Based on the relationship between $\delta^{15}$N–NO$_x$ values and emission regulations, this study evaluated the temporal variation of $\delta^{15}$N–NO$_x$ emissions from ships larger than 300 GT in the international merchant fleet by establishing a mass-weighted model. With the implementation of stricter emission standards, the $\delta^{15}$N–NO$_x$ values of ship emissions continued to rise and the growth rate increased from 0.25‰/year (2001–2012) to 0.40‰/year (2013–2021). Assuming the same age distribution of vessels after 2022 as in 2021, it can be estimated that the $\delta^{15}$N–NO$_x$ values will continue to increase before 2040 (0.40‰/year) and will remain basically flat after 2046 when all ships will meet the Tier III emission standard.

While the $\delta^{15}$N–NO$_x$ values obtained in this study may not provide an exact representation of emissions from all ship types, they are valuable for assessing source characterization of traffic exhaust in terms of $\delta^{15}$N–NO$_x$. Meanwhile, the $\delta^{15}$N–NO$_x$ values calculated through mass weighting, considering the age distribution of ships, exhibit a narrower range of variation and hold greater practical significance. In addition, local $\delta^{15}$N–NO$_x$ values can be calculated by incorporating the age distribution and emission reduction level of ships in the study area. This would facilitate a more accurate contribution of ship emissions to atmospheric NO$_x$, and impact on NO$_3^-$ air quality and nitrogen deposition.

**Data availability.** The $\delta^{15}$N–NO$_x$ values emitted from cars fueled by diesel, gasoline and LPG used in this paper can be found in researches by Walters et al (Walters et al., 2015a; Walters et al., 2015b). The age distribution of ships larger than 300 GT in the international merchant fleet during 2001 and 2021 used to develop the mass-weighted model of $\delta^{15}$N–NO$_x$ values emitted from ships are obtained from a series of reviews and prospects of world shipping market from 2001 to 2021 published in *Ship & Boat* (Yang and Zhou, 2002; Zhou et al., 2004; Meng et al., 2005; Meng and Huang, 2006; Meng et al., 2007;

Qi et al., 2008; Qin et al., 2009; Qi et al., 2010; Li et al., 2011; Qin et al., 2012; Qi et al., 2013; Li et al., 2014; Qin et al., 2015; Li et al., 2016; Qin et al., 2017, 2018; Shen and Qi, 2019; Qi et al., 2020; Liu et al., 2021; Wei et al., 2022). Corresponding data for the collected ship-emitted $NO_x$ samples can be accessed on request to the corresponding author (Chongguo Tian, cgtian@yic.ac.cn).

**Author contributions.** The manuscript was written through contributions of all authors. ZS, ZZ, CT and FZ designed the research; ZS, ZZ and ZL conducted the sample collection; ZS and YT performed the chemical analyses; ZS and CT analyzed the data, carried out the simulations and wrote the original article; RS, YC, JL and GZ helped with article submissions. All authors have given approval to the final version of the manuscript.

**Competing interests.** The contact author has declared that none of the authors has any competing interests.

**Acknowledgements.** This research was financially supported by the National Natural Science Foundation of China (nos. 41977190) and the NSFC–Shandong Joint Fund (nos. U1906215).

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
