# Peer review of "Characterization of the Nitrogen Stable Isotope Composition ( $\delta^{15}$ N) of Ship-emitted NOx"

_EGUsphere, 2023_

## Author Comment (AC1)

**Response to the Reviewers' Comments and Suggestions — egusphere-2023-500**

Dear editorial board of Atmospheric Chemistry and Physics,

We are very grateful to all the reviewers for the beneficial comments and suggestions. We have revised this manuscript accordingly. Please find below our point-by-point responses (blue) to each reviewer's comments (black).

Sincerely yours,

Dr. Chongguo Tian

Yantai Institute of Coastal Zone Research, Chinese Academy of Sciences, Yantai, 264003, China

Email: cgtian@yic.ac.cn

Tel: 86-535-2109160

Fax: 86-535-2109000

**Reviewer 1 (Page 2–12)**

Sun and co-authors measure for the first time, the nitrogen stable isotopic composition of ship emitted $NO_x$ ($\delta^{15}N$–$NO_x$). They find a large range in values of ship emitted $\delta^{15}N$–$NO_x$ that increased with tightening emission regulations and reduced atmospheric $NO_x$ concentrations. Emission regulations are found to have the greatest influence on ship emitted $\delta^{15}N$–$NO_x$, which was explored using multiple statistical techniques. The biggest difference in ship emitted $\delta^{15}N$–$NO_x$ occurring when IMO Tier III emission standards where implemented, and ships began adopting $NO_x$ emission control technologies like selective catalytic reduction (SCR).

With transportation fast becoming one of the most important emission sources of anthropogenic $NO_x$, reliable values of ship emitted $\delta^{15}N$–$NO_x$ are essential to accurate assessments of atmospheric $NO_x$ sources, especially in coastal regions. A such, these data are valuable and should be considered for publication in ACP. While the data presented support the main conclusions (barring the erroneous Fig. 2), some additional information particularly regarding the implications of the data (Sect. 3.3) may benefit readers when interpreting the results, and those who intend to build upon this work for future studies.

**Response and Revisions:** We appreciate the insightful advice from the reviewer and have revised and supplemented the manuscript accordingly. Please find our detailed responses below. Thanks again for the valuable feedback.

**Major revisions stated below:**

Figure 2 — Vessel category as opposed to emission regulation stages are shown on the x-axis. The

means reported do not match the symbols (red squares). It appears the wrong categories are plotted here.

**Response and Revisions:** We sincerely apologize for the incorrect categorization in Figure 2 in the previous manuscript. Following the reviewer's opinion, we have modified it to correctly display $\delta^{15}N$ values in $NO_x$ emitted from ships under different emission regulations established by the IMO. Please refer to the updated figure presented below.

[Figure]

**Figure 2.** $\delta^{15}N–NO_x$ values emitted from ships under different emission regulations established by the IMO. (red square, mean; center line, median; box limits, upper and lower quartiles; whiskers, 1.5 times interquartile range; points, outliers (pink, cruising; green, hoteling; blue, maneuvering); outer line, distribution of data). Mean ± standard deviation of $\delta^{15}N–NO_x$ values of each group is marked on the bottom of the panel. The difference and *p* values indicating the distinction between two groups are marked on the upper of the panel (the Mann–Whitney U test).

Figure S9 – What do the yellow and dark blue sections of the 100 % bar chart represent? Additionally, it is unclear how the TSi was calculated based on the age distribution of ships during 2001 and 2021. Is there a maximum age beyond which ships no longer comply with Tier I, II or III emission regulations? Or is the age of the ship indicative of which emission standard it complies with? Clarification is needed.

**Response and Revisions:** The yellow and light blue sections represent the proportions of ships that are less than or equal to 20 years old and older than 20 years, respectively. The green line graph represents the ship stock. The legend of Fig S9 was not shown completely, and we have therefore corrected it. The modified Figure S9 (as Figure S8 in the revised manuscript) in the manuscript is shown below.

[Figure]

**Figure S8.** The age distribution of ships larger than 300 gross tonnage (GT) in the international merchant fleet during 2001 and 2021.

$TS_i$ represents the number of vessels complying with the i-th emission regulation, where the subscript i ranges from 1 to 4, representing before Tier I, Tier I, Tier II, and Tier III, respectively. To limit $NO_x$ emissions in the exhaust gas of marine diesel engines, the International Maritime Organization (IMO) provides clear regulations in Annex VI of the International Convention for the Prevention of Pollution from Ships (MARPOL) (as shown in Table C1, https://dieselnet.com/standards/inter/imo.php#other). Since January 1, 2000, newly constructed or significantly refurbished marine diesel engines are required to meet the requirements of Tier I standard. Subsequently, stricter standards such as Tier II and Tier III have been gradually introduced to further reduce the environmental impact of ship emissions. The IMO Tier III standard among them applies only to ships operating within the Nitrogen Oxide Emission Control Areas (NECAs), which mainly include the Baltic Sea area, the North American Emission Control Area, and the Caribbean Sea Emission Control Area.

**Table C1.** MARPOL Annex VI $NO_x$ Emission Limits (g/kWh).

| emission regulation | effective date | rated speed/N (rpm) | | |
|---|---|---|---|---|
| | | N < 130 | 130 ≤ N < 2000 | N ≥ 2000 |
| Tier I | 2000/01/01 | 17.0 | $45N^{-0.2}$ | 9.8 |
| Tier II | 2011/01/01 | 14.4 | $44N^{-0.23}$ | 7.7 |
| Tier III (in NECAs) | 2016/01/01 | 3.4 | $9N^{-0.2}$ | 1.96 |

Therefore, the age of a ship can indicate the emission standard it complies with. Ships built before January 1, 2000, are not subject to specific emission standards; ships built between January 1, 2000,

and January 1, 2011, comply with Tier I; ships built between January 1, 2011, and January 1, 2016, comply with Tier II; ships built on or after January 1, 2016, must comply with Tier III if operating within the NECAs. The annual $TS_i$ can accordingly be calculated based on the distribution of ship ages.

In consideration of the reviewer's concerns, we have supplemented the details of emission regulations for ship-emitted $NO_x$ prescribed by IMO in Table 3 of the revised manuscript. Besides, the correlation between emission standards met by ships and ship age was also added in Section 3.2 on page 10–11, line 313–317: "The reason for considering compliance with the emission regulation for ship engines as a criterion is that the IMO emission regulation is the most important measure to restrict $NO_x$ emissions from ships and once each standard was proposed by the IMO, newly constructed or significantly refurbished marine diesel engines must comply with its requirements. Therefore, the emission regulation that a ship complies with is actually related to its age."

**Minor revisions stated below:**

Some introductory text regarding the utilisation of stable isotope ratios is required, i.e., a description of delta notation and the units per mil (‰).

**Response and Revisions:** As suggested by the reviewer, the reporting unit for stable isotope ratios has been added in the revised Section 2.2 titled "Chemical and isotopic analysis" (page 7, line 181–185)."Finally, the $\delta^{15}N$ of the produced $N_2O$ was analyzed by an isotope ratio mass spectrometer (MAT253, Thermo Fisher Scientific, Waltham, MA) and the $\delta^{15}N$ values were reported in parts per thousand relative to the international standards (IAEA–NO–3, USGS32, USGS34, and USGS35) (Bohlke et al., 2003):

$$\delta^{15}N = \left[ \frac{\left(^{15}N/^{14}N\right)_{sample}}{\left(^{15}N/^{14}N\right)_{standard}} - 1 \right] \times 1000 \qquad (1)"$$

Line 36: Define $NO_x$ (NO + $NO_2$)

**Response and Revisions:** Following the reviewer's opinion, the sentence has been modified as "Due to its detrimental impact on the environment, ecology, and public health, anthropogenic emissions of nitrogen oxides ($NO_x$), including nitric oxide (NO) and nitrogen dioxide ($NO_2$), have drawn considerable attention in recent decades." Page 2, Line 41–43

Line 37: Replace vital with important

**Response and Revisions:** As recommended by the reviewer, we have made the corresponding modifications. Page 2, Line 44

Line 50: in comparison with the year 2000

**Response and Revisions:** In response to the reviewer's comment, we have changed the phrase "levels of" to "year" in the revised manuscript. Page 2, Line 56–57

Line 54 to 56: A concluding remark is required here. Something like, the current world merchant fleet thus comprises of more, newly built vessels that have benefitted from the implementation of

emission reduction technologies.

**Response and Revisions:** In response to the reviewer's suggestion, we have added the summarizing transitional sentence "The current world merchant fleet thus comprises more newly built vessels that have benefited from the implementation of emission reduction technologies." in the revised manuscript to enhance the overall coherence of the logic. Page 2–3, Line 64–65

Line 70: Primary contributor to what? Total anthropogenic $NO_x$?

**Response and Revisions:** The phrase "the primary contributor" refers to the main contributor to $NO_x$ emissions from ships. In order to make the expression clearer, we further examined the references here and modified the sentence to "Simulations of atmospheric air quality models further revealed that the intensive emissions from ships within 12 Nm from the coastline were the primary contributor to ship-related $NO_x$, accounting for approximately 70 % of the total emissions (Wang et al., 2018)." Page 3, Line 80–83

Line 71: Rather than state three major urban agglomerations, name the regions.

**Response and Revisions:** In order to make the expression clearer, we further examined the references here and modified the sentence to "Simulations of atmospheric air quality models further revealed that the intensive emissions from ships within 12 Nm from the coastline were the primary contributor to ship-related $NO_x$, accounting for approximately 70 % of the total emissions (Wang et al., 2018)." Page 3, Line 80–83

Line 74: Replace the phrase "in the face of", with "due to"

**Response and Revisions:** In accordance with the advice of the reviewer, inappropriate expressions have been replaced. Page 3, Line 86

Line 77: Is a powerful method used to apportion

**Response and Revisions:** Accordingly, we have changed the phrase "is one of the powerful methods" to "is a powerful method" in the revised manuscript. Page 3, Line 89

Line 87: To address the lack of $\delta^{15}N–NO_x$ measurements associated with ship emissions, rather than "Aiming at the knotty problem of lacking $\delta^{15}N–NO_x$".

**Response and Revisions:** We made corresponding modifications based on the suggestion of the reviewer. Page 3, Line 99–100

Line 110: Some description of how changes in meteorological conditions may affect measured $\delta^{15}N–NO_x$ emitted from ships, or some references to previous work that has explored this would be beneficial so that the reader is aware of any potential implications. This should be done before noting that the effect is beyond the scope of the study, and will not be addressed further.

**Response and Revisions:** Meteorological conditions primarily affect ship emissions by influencing the operational mode of the ship. For example, sailing with a tailwind or headwind can result in variations in the engine load, thereby altering the $\delta^{15}N–NO_x$ values from ship emissions. Quantifying the exact extent of these changes may pose challenges. Ship speed indicating workloads of ship engines is one of key factors to identify different ship activities. Duan et al. (2022) used a

vector model incorporating ship-measured speed, wind speed, wave height, and water flow velocity to correct ship speed. The results indicated that when the average wind speed, wave height, and ocean current were relatively high, specifically when the average wind speed exceeded 5.5 m s$^{-1}$, the average corrected speed only accounted for 5.88 % to 7.69 % of the measured speed, meaning that the impact on the engine workload of the ship was also minimal.

In order to minimize the impact of meteorological conditions on the test results, we picked calm weather (with zephyrs and gentle waves) and appropriate temperature to carry out ship experiments as presented in Table S1 of SI. In conclusion, similar to some previous researches regarding NO$_x$ emitted from ships (Wang et al., 2019; Jiang et al., 2019; Zhang et al., 2018; Zhao et al., 2020), we did not take into account the influence of meteorological conditions during sampling, considering its minimal impact and difficulty in quantification. The final results also indicated that the operational status of the ship was not the most significant factor affecting the emission of $\delta^{15}$N–NO$_x$ values from ships.

We have included the necessary explanation and relevant references regarding this in the revised manuscript: "Meteorological conditions primarily affect ship emissions by influencing the operational mode of the ship, and this impact is relatively small and difficult to quantify (Huang et al., 2018; Duan et al., 2022; Zhao et al., 2020). Considering the unpredictable weather circumstances, we attempted to pick calm weather (with zephyrs and gentle waves) and appropriate temperature to carry out ship experiments. As presented in Table S1 of the Supporting Information (SI), the temperature, wind speed and relative humidity ranged from 1 to 27°C, from 2.8 to 5.1 m s$^{-1}$, and from 49 to 68 % during the observation period, respectively, which were obtained from the local weather station established by the China Meteorological Administration (Wang et al., 2019). Consequently, similar to some previous researches regarding NO$_x$ emitted from ships (Wang et al., 2019; Jiang et al., 2019; Zhang et al., 2018; Zhao et al., 2020), the influence of meteorological conditions during sampling was not taken into account in this study." Page 4, Line 121–131

Line 116: but boilers were not sampled since the contribution of boiler exhaust to NO$_x$ emissions is weak.

**Response and Revisions:** As recommended by the reviewer, corrections have been made to the grammar mistakes and inappropriate expressions in that sentence. "If the ship also has auxiliary engines (AE) and boilers, NO$_x$ emitted from AE were also collected, but boilers were not sampled since these make a small contribution to NO$_x$ emissions, less than 5% compared to those from ME and AE (Wan et al., 2020; Shi et al., 2020; Chen et al., 2017)." Page 6, Line 136–139

Line 130: How was isotopic fractionation avoided?

**Response and Revisions:** Throughout the sampling process, to avoid the occurrence of $\delta^{15}$N fractionation, each sampling process and analyzing process were conducted carefully and exactly. The connecting pipes used were stainless steel bellows (1.5 m or 3.0 m with an inner diameter of 40 mm) and Teflon tubes (1.5 m with an inner diameter of 12.77 mm); and the exhaust gas was

pumped at a flow rate of $1.0\ \text{L min}^{-1}$ into the gas-washing bottle. Therefore, the time for which $NO_x$ were present in the pipe was less than 4 min. This was much lower than the average lifetime of NO in the air ($\sim 15216.3$ s) (Massman, 1998); the fractional distillation could thus be ignored. We apologize for the previous calculation error and have made the necessary corrections in the revised manuscript: "The total length of the connecting pipe (stainless steel bellows and Teflon tubes) from the ship chimneys to gas-washing bottles was about 2 m or 4 m for different ships and $NO_x$ were accordingly present in the pipe for less than 4 min, which was significantly shorter than the normal airborne NO lifespan ($\sim 15216.3$ s), meaning that fractional distillation could be ignored (Massman, 1998)." Page 6, Line 161–165

Line 197: and $NO_x$ produced thermally

**Response and Revisions:** Following the reviewer's opinion, we have added "$NO_x$" before the phrase "produced thermally by internal combustion engines tends to be...". Page 9, Line 262

Line 212: suggesting

**Response and Revisions:** In order to make the expression clearer, we modified the sentence to "Overall, the negative $\delta^{15}N–NO_x$ values emitted from ships in this research, with minimal influence from fuels, suggest that these values are more likely associated with the production of thermal $NO_x$ rather than the conversion of nitrogen in the fuels (Walters et al., 2015a; Walters et al., 2015b; Zong et al., 2022)." Page 9, Line 276–279

Line 223: Refer to Fig. 1 at the end of the sentence.

**Response and Revisions:** As recommended by the reviewer, corrections have been made to make sentences more clearly directed. Page 10, Line 288

Line 224 to 226: I found this concluding sentence slightly confusing a little vague. Do you mean that given ship emitted $\delta^{15}N–NO_x$ differs substantially to the $\delta^{15}N–NO_x$ produced by diesel, gasoline and LPG powered combustion engines, further investigation is required to determine the factors influencing ship emitted $\delta^{15}N–NO_x$ for the accuracy of source apportionment etc. If so, please describe as such, alternatively please clarify the concluding sentence here.

**Response and Revisions:** We want to express the exact meaning as stated by the reviewer. In order to make the conclusion clearer and strengthen its logical connection with the previous description, we have rephrased this section as follows: "The significant difference observed highlights the importance of addressing the data gap in measuring $\delta^{15}N–NO_x$ emissions from ships for accurately apportioning atmospheric $NO_x$ sources in coastal zones, especially in some port areas with high ship activity. Further discussion is needed to understand the reasons behind the noticeable difference and to better utilize the measured values of ship-emitted $\delta^{15}N–NO_x$, including the variations in $\delta^{15}N–NO_x$ values and their primary influencing factors." Page 10, Line 288–293

Figure 1: For figure one, there is no need to include the word "values" in the y-axis label. The colors of each box and whisker diagram could cause confusion, as I originally assumed like colors were related in some way (i.e., similar types). If you are not able to use a different color for each type,

please consider having all ship types (Ship, Ship 6.1 and Ship 8.7) in the same color, all gasoline type (gasoline and gasoline 5.2) etc. in the same colors. It would also be useful to indicate on the face of the figure, which types have been adjusted for not having SCR, perhaps with square brackets above or beneath the relevant box and whisker diagrams.

**Response and Revisions:** According to the reviewer's suggestions, we made modifications to Figure 1, including removing the word "values" from the y-axis label and changing the $\delta^{15}N–NO_x$ values emitted by each type of engine to the same color. Moreover, we adjusted the colors of the modified values to be lighter in shade. The $\delta^{15}N–NO_x$ values emitted by gasoline vehicles, diesel vehicles, and ships without $NO_x$ catalytic reduction devices were corrected using enrichment factors, as described on page 15, line 428–436 in the revised manuscript. Please refer to the updated Figure 1 in the manuscript presented below.

[Figure]

**Figure 1.** $\delta^{15}N–NO_x$ values emitted from ships in this study and cars fueled by diesel, gasoline and liquefied petroleum gas (LPG) reported from other references (Walters et al., 2015a; Walters et al., 2015b). The Ship (6.1) and Ship (8.7) indicate that $\delta^{15}N–NO_x$ values from ships without selective catalytic reduction (SCR) systems are corrected by 6.1 ‰ and 8.7 ‰, respectively. The Diesel (C) and Gasoline (5.2) indicates that $\delta^{15}N–NO_x$ values from diesel and gasoline cars without SCR systems are corrected using 6.1 ‰ or 8.7 ‰, and 5.2 ‰, respectively. (red square, mean; center line, median; box bounds, upper and lower quartiles; whiskers, 1.5 times interquartile range; points, outliers; outer line, data distribution). The $p$ values indicating the distinction between two groups are marked on the upper of the panel (the Mann–Whitney U test).

Line 246: These classification indicators are considered because, as opposed to "The consideration is because".

**Response and Revisions:** We apologize for the lack of clarity in the previous manuscript. For the purpose of sidestepping any confusion, we have rephrased the reasons for selecting these

influencing factors: "The reason for considering compliance with the emission regulation for ship engines as a criterion is that the IMO emission regulation is the most important measure to restrict $NO_x$ emissions from ships and once each standard was proposed by the IMO, newly constructed or significantly refurbished marine diesel engines must comply with its requirements. Therefore, the emission regulation that a ship complies with is actually related to its age. Besides, the category, fuel type, and actual operational status are often used to assess the variation in $\delta^{15}N–NO_x$ values from vehicles (Walters et al., 2015a; Walters et al., 2015b)." Page 10–11, Line 313–319

Line 254: Instead of (small p values), (indicated by p values < xxx) would be more descriptive.

**Response and Revisions:** As shown in Figure 2 and Figure S3, we performed the Mann–Whitney U test to examine the significance of differences in $\delta^{15}N–NO_x$ values among emission standards and ship categories, respectively. The p-values for each group under the respective indicator were given in the figure and most of them were below 0.001, indicating a significant difference. However, there were exceptions to this, such as the difference between $\delta^{15}N–NO_x$ values emitted from ships meeting Tier I and Tier II emission regulations (0.16), and the difference between $\delta^{15}N–NO_x$ emissions from passenger ships and research vessels (0.86). Therefore, the term "small $p$ values" was used to summarily demonstrate that the emission regulation and the ship category are the primary influencing factors for ship-emitted $\delta^{15}N–NO_x$ values.

Line 255: By "divided by" do you mean between the two indicators?

**Response and Revisions:** We are very sorry for the inappropriate wording in the previous manuscript. What we meant to express is that there are differences in $\delta^{15}N–NO_x$ values among different groups of ships after categorizing them based on each indicator. To prevent any misunderstandings, we have modified the sentence to "Similar significant differences (small $p$ values) in $\delta^{15}N–NO_x$ values between types of emission regulations, and different vessel categories were calculated by the Mann–Whitney U test, as displayed in Fig. 2 and Fig. S3, respectively." Page 11, Line 325–327

Line 254 to 256: This sentence is slightly confusing, do you mean: Similar significant differences (small $p$ values) in $\delta^{15}N–NO_x$ values between types of emission regulations, and different vessel categories were calculated by the Mann–Whitney U test, as shown in Fig. 2 and Fig. S2, respectively. If so, please clarify.

**Response and Revisions:** Our intended meaning is consistent with the reviewer's opinions. Based on this, we have made corrections in the revised manuscript. Page 11, Line 325–327

Line 283 to 285: It is unclear to me what great discrepancy is being referred to here. Please clarify.

**Response and Revisions:** In this paragraph, following the statement "The largest average growth rate of $\delta^{15}N–NO_x$ values emitted from ships between the three adjacent phases from 1 to 4 occurred between the implementation of IMO Tier II and III (stage 3–4, 54.1 %), followed by before and after the implementation of Tier I (stage 1–2, 36.4 %), and between Tier I and II (stage 2–3, 17.8 %)" (page 12, line 363–366), we explained the reasons for the differences in $\delta^{15}N–NO_x$ values emitted

from ships between two adjacent regulatory stages in chronological order. The "great discrepancy" here refers to the average growth rate of $\delta^{15}$N–NO$_x$ values emitted from ships before and after the implementation of Tier I (stage 1–2, 36.4 %).

Given that fuel optimization technologies and precombustion control technologies are two main control methods to meet Tier I and Tier II, we explained the reasons why the implementation of Tier I and Tier II resulted in increased ship-emitted $\delta^{15}$N–NO$_x$ values, as well as the reason why the average growth rate of $\delta^{15}$N–NO$_x$ values between Tier I and Tier II is lower than that before and after the implementation of Tier I in the revised manuscript: "Since the implementation of IMO Tier I on January 1, 2000, fuel optimization technologies (fuel emulsification, fuel desulfurization and fuel additives) and pre-combustion control technologies (fuel injection strategy, water injection strategy, Miller cycle, two-stage turbocharging, and dual-fuel combustion strategy), as demonstrated in Table 3, have been applied and experienced rapid development. The former reduce the production of NO$_x$ by changing the composition of fuel or adding additives to improve the combustion process, while the latter primarily suppress the formation of NO$_x$ by modifying the internal structure of diesel engines or adjusting parameters (Ampah et al., 2021; Deng et al., 2021; Lion et al., 2020). The average growth rate of $\delta^{15}$N–NO$_x$ values between Tier I and Tier II (17.8 %) is lower than that before and after the implementation of Tier I (36.4 %). This difference can be attributed to the fact that different emission standards have varying levels of impact on NO$_x$ emission reductions. Compared to the optimization and development of emission reduction technologies between Tier I and II, the implementation of Tier I fills the gap in ship engine and fuel emission reduction measures, which clearly has a greater impact on NO$_x$ emissions." Page 12–13, Line 366–379

Line 287: Instead of "The insignificant discrepancy of the $\delta^{15}$N–NO$_x$ values from ships implementing Tier I and Tier II", "The insignificant difference between $\delta^{15}$N–NO$_x$ values from ships implementing Tier I and Tier II" is clearer.

**Response and Revisions:** We are very sorry for the unclear expression in the previous manuscript. After careful consideration, we have rephrased the sentence as "The average growth rate of $\delta^{15}$N–NO$_x$ values between Tier I and Tier II (17.8 %) is lower than that before and after the implementation of Tier I (36.4 %)." Page 12, Line 373–375

Line 304: Please refer to Fig. 1 at the end of the sentence here.

**Response and Revisions:** As recommended by the reviewer, corrections have been made to make sentences more clearly directed. Page 13, Line 400

Line 319: Please clarify "The discrepancy", i.e., "The stronger relationship between $\delta^{15}$N–NO$_x$ and NO$_x$ concentration for vehicles with NO$_x$ emission control technologies versus vehicles without, is attributed to the enrichment of $\delta^{15}$N.."

**Response and Revisions:** According to the reviewer's suggestion, we have clarified "The discrepancy" here. Page 14, Line 418–420

Section 3.2 would benefit from a table summarizing the 4 emissions regulation stages, whether Tier

I, II or III was implemented, what the major differences are (e.g., SCR/ fuel optimisation and precombustion control technologies), the corresponding $\delta^{15}N–NO_x$ and $NO_x$ concentrations, and emission factors of $NO_x$ pertaining to Section 3.3. This will also be helpful to Section 3.3 to summarise the parameters set for the mass weighted model (i.e, emission factors and $\delta^{15}N–NO_x$ values).

**Response and Revisions:** We are grateful for the recommendation put forward by the reviewer. The table which provides a summary of emission regulations for ship-emitted $NO_x$ at each stage is indeed extremely helpful in clarifying the significant impact of emission regulations on $\delta^{15}N–NO_x$ values emitted from ships. Therefore, we have summarized the information of each stage, including the effective date of the emission regulation, limits for $NO_x$ emission factors (g/kWh), $NO_x$ emission reduction technologies, and the measured $NO_x$ concentrations (ppm) and $\delta^{15}N–NO_x$ values (‰) in our study, in Table 3, and it has been supplemented in the revised manuscript.

**Table 3.** Summary of ship-emitted $NO_x$ at each stage of emission regulations.

| stage | effective date of the emission regulation | $NO_x$ emission limits (g (KWh)$^{-1}$) | | | $NO_x$ emission reduction technologies | $NO_x$ (ppm) | $\delta^{15}N–NO_x$ (‰) |
|---|---|---|---|---|---|---|---|
| | | $N^a < 130$ | $130 \leq N < 2000$ | $N \geq 2000$ | | | |
| Before Tier I | | | | | | $96.6 \pm 28.6$ | $-33.8 \pm 1.83$ |
| Tier I | 2000/01/01 | 17.0 | $45N^{-0.2}$ | 9.8 | fuel optimization: fuel emulsification, fuel desulfurization, fuel additives; pre-combustion control technologies: fuel injection, water injection, Miller cycle, two-stage turbocharging, dual-fuel combustion | $239 \pm 93.0$ | $-21.5 \pm 6.67$ |
| Tier II | 2011/01/01 | 14.4 | $44N^{-0.23}$ | 7.7 | | $169 \pm 156$ | $-17.8 \pm 9.88$ |
| Tier III (in NECAs$^b$) | 2016/01/01 | 3.4 | $9N^{-0.2}$ | 1.96 | exhaust after-treatment system: exhaust gas recirculation (EGR), selective catalytic reduction (SCR), non-thermal plasma (NTP), seawater flue gas desulphurization (SWFGD) | $122 \pm 91.6$ | $-8.12 \pm 8.84$ |

[a] N: rated speed (rpm)

[b] NECAs: the Nitrogen Oxide Emission Control Areas, mainly including the Baltic Sea area, the North American Emission Control Area, and the Caribbean Sea Emission Control Area.

Line 381: Please include a description of units either here, or in the table.

**Response and Revisions:** Grams per kilowatt-hour (g (KWh)$^{-1}$) is a commonly used unit of emission factors (EF) to measure the emissions of pollutants produced per unit of energy consumed. In this unit, grams represent the mass of the emitted pollutants, while kilowatt-hours represent the unit of energy consumed. The g (KWh)$^{-1}$ value indicates the amount of pollutants emitted per kilowatt-hour of energy consumed. A lower value indicates lower emissions of pollutants per unit of energy consumed, indicating a cleaner and more environmentally friendly combustion process.

We have supplemented the definition of EF and its unit, g (KWh)$^{-1}$, in the first mention of the emission factor in the introductory section ("1 Introduction") of the text: "The emission factor (EF)

of a ship refers to the parameter that measures the rate or proportion of pollutants generated by the ship during its operation, usually expressed in grams per kilowatt-hour (g $(KWh)^{-1}$). The g $(KWh)^{-1}$ value indicates the number of pollutants emitted per kilowatt-hour of energy consumed. A lower EF value indicates lower emissions of pollutants per unit of energy consumed, indicating a cleaner and more environmentally friendly combustion process." Page 3, Line 67–71

**Reviewer 2 (Page 12–28)**

Review of Sun et al.

This work reports on measurements of the isotope composition of $NO_x$ emitted from ships under a variety of operating mechanisms, fuel types, ship types/ages, etc. The results are reported both in terms of the direct observations as well as in the context of weighting by emissions and correcting for fractionations associated with scrubbing technology. The study also predicts the changes in the isotope composition of ship emissions over time, which is very useful for future studies. It is a very well-done study and overall, the results are compelling. The comments overall are minor, but important to the scientific merit of the work.

The manuscript appears written for a short form journal and would benefit from a fuller discussion of implications and a conclusions section. There also several points recommended below to pull in text and/or figures from the supplement since there is space to do so for ACP. Most of the comments surround clarification of weighting specifics for calculating the final isotopic composition, making recommendations on how best to use the reported values for future studies and clarifying meaning of text in several places.

**Response and Revisions:** We appreciate the thought-provoking guidance from the reviewer. Accordingly, we have made necessary revisions and additions to the manuscript, including providing a more detailed discussion of the research findings, moving several contents from the Supplementary Information (SI) to the main text and so on. For details, please refer to the follow-up replies. Thanks again for these professional comments.

**Title:**

I appreciate the "first measurements" specificity here, but I think the authors do more than just measure the $d^{15}N$. I would suggest using the word "characterization" and also making it clear that these values apply to emissions. For example: "Characterization of the nitrogen stable isotope composition ($d^{15}N$) of ship $NO_x$ emissions"

**Response and Revisions:** We appreciate the valuable advice from the reviewer and changed the title of this paper to "Characterization of the Nitrogen Stable Isotope Composition ($\delta^{15}N$) of Ship-emitted $NO_x$".

**Abstract:**

The abstract is a good representation of the study. It would be helpful to clarify the values presented. Are these the directly observed values or weighted values, specify this. At the end of the abstract, it would also be good to acknowledge that you present a framework to compute accurate assessments over time.

**Response and Revisions:** The $\delta^{15}N$–$NO_x$ values mentioned in the abstract are all weighted values, so we have noted them in the revised manuscript as suggested by the reviewers: "Results showed that $\delta^{15}N$–$NO_x$ values from ships, which were calculated by weighting the emission values from the main engine and auxiliary engine of the vessel, ranged from −35.8 ‰ to 2.04 ‰ with a mean ± standard deviation of −18.5 ± 10.9 ‰." Page 1, Line 25–28

In addition, we rephrased the last part of the abstract to acknowledge that we present a framework to compute accurate assessments over time: "Based on the relationship between $\delta^{15}N$–$NO_x$ values and emission regulations observed in this investigation, a mass-weighted model to compute accurate assessments over time was developed and the temporal variation in $\delta^{15}N$–$NO_x$ values from ship emissions in the international merchant fleet was evaluated." Page 1, Line 32–35

**Introduction:**

(1) In the introduction, it would be useful to give more background information about how much $NO_x$ is emitted under different operating conditions (i.e. why are three operating conditions specifically chosen, how are they regulated, etc). (2) As well as a clearer description or table of the emissions standards – i.e., Tier I, Tier II, Tier III. I was not familiar with much of the policy and regulatory framing before reading this paper. (3) Additionally, it would be helpful on line 50 to define MARPOL and give a bit more information in a table about the emissions regulations and on which dates these changed (the predicted time series in Figure 3 is based on this).

**Response and Revisions:** Because this comment contains multiple small questions, we have divided it according to the meanings, as the question (1) to (3). Accordingly, we will reply to them in division as follows.

(1) We agree with the reviewer's perspective that providing more background information on different operating conditions of ships is necessary. Therefore, in Section 2.1, "Sampling campaign," we have added the necessity of collecting $NO_x$ samples from ships under different operating conditions and provided details on how the operating conditions were categorized. "Under different operating conditions, ships have different speeds and loads, and factors such as fuel consumption rate, combustion temperature and time will change, which in turn affects the amount of $NO_x$ emitted from ships (Liu et al., 2022). $NO_x$ samples were collected under actual operating conditions of ships, and actual ship speed was monitored by the global positioning system (GPS) equipped on board. Samples were grouped based on three operating modes on the basis of the actual speed of each ship: cruising mode (> 8 knots, ship operating at higher speed), maneuvering mode (1–8 knots, ship operating at lower speed when approaching berths or anchorages)and hoteling mode (< 1 knot, ship

(2) We are grateful for the recommendation put forward by the reviewer. We have summarized emission regulations for ship-emitted $NO_x$ at each stage as Table 3 and included it in the revised manuscript. This table includes the effective date of the emission regulation, limits for $NO_x$ emission factors (g/kWh), $NO_x$ emission reduction technologies, and the measured $NO_x$ concentrations (ppm) and $\delta^{15}N$–$NO_x$ values (‰) in our study.

**Table 3.** Summary of ship-emitted $NO_x$ at each stage of emission regulations.

| stage | effective date of the emission regulation | $NO_x$ emission limits (g (KWh)$^{-1}$) | | | $NO_x$ emission reduction technologies | $NO_x$ (ppm) | $\delta^{15}N$– $NO_x$ (‰) |
|---|---|---|---|---|---|---|---|
| | | $N^a <$ 130 | $130 \leq N$ < 2000 | $N \geq$ 2000 | | | |
| Before Tier I | | | | | | 96.6 ± 28.6 | −33.8 ± 1.83 |
| Tier I | 2000/01/01 | 17.0 | $45N^{-0.2}$ | 9.8 | fuel optimization: fuel emulsification, fuel desulfurization, fuel additives; pre-combustion control technologies: fuel injection, water injection, Miller cycle, two-stage turbocharging, dual-fuel combustion | 239 ± 93.0 | −21.5 ± 6.67 |
| Tier II | 2011/01/01 | 14.4 | $44N^{-0.23}$ | 7.7 | | 169 ± 156 | −17.8 ± 9.88 |
| Tier III (in NECAs[b]) | 2016/01/01 | 3.4 | $9N^{-0.2}$ | 1.96 | exhaust after-treatment system: exhaust gas recirculation (EGR), selective catalytic reduction (SCR), non-thermal plasma (NTP), seawater flue gas desulphurization (SWFGD) | 122 ± 91.6 | −8.12 ± 8.84 |

[a] N: rated speed (rpm)

[b] NECAs: the Nitrogen Oxide Emission Control Areas, mainly including the Baltic Sea area, the North American Emission Control Area, and the Caribbean Sea Emission Control Area.

(3) We have provided a definition of "MARPOL" the first time it is mentioned in the paper (page 2, line 51–54). "For instance, the International Convention for the Prevention of Pollution from Ships (MARPOL) implemented by the International Maritime Organization (IMO) is a momentous international treaty concerning the avoidance of pollution generated by vessels during operations or from unexpected causes." And the Table 3, which has been added to the revised manuscript as indicated in the answer to question (2) above, presents more information about the emission regulations for ship-emitted $NO_x$ at each stage.

**Methodology:**

It would be helpful to have a photo or diagram of the set up on the ship (this could go in the supplement). It is difficult to picture the setup based on the limited description here.

**Response and Revisions:** According to the reviewer, we have supplemented a photo taken during sampling on the Y2 ship in the Supplementary Information (Figure S1). This photo illustrates the setup of the on-board sampling device, with the yellow arrow indicating the emission of exhaust from the ship, and the circular marked device representing filters.

[Figure]

**Figure S1:** The set up on the ship during sampling. The yellow arrow indicates the emission of exhaust from the ship.

(1) On line 112 it is stated that any impact of meteorological conditions is ignored. I suspect this fine given that the sampling is taking place within the smokestack. However, could the authors provide evidence for a lack of systematic differences for summer vs winter conditions for instance? (2) This will come up again later, but there is little discussion of the internal variability of the signals within an operating condition (e.g., operating under cruising mode produces a large range of values for the same ship emissions) and it is curious whether the emissions change at all within a specific operating condition due to other factors such as meteorology or the respond of the engine to outside temperature conditions.

**Response and Revisions:** We have segmented this comment as question (1) and (2) according to their meanings and provided responses individually in accordance with this division.

(1) Meteorological conditions primarily affect ship emissions by influencing the operational mode of the ship. For example, sailing with a tailwind or headwind can result in variations in the engine load, thereby altering the $\delta^{15}N$–$NO_x$ values from ship emissions. Quantifying the exact extent of these changes may pose challenges. Ship speed indicating workloads of ship engines is one of key factors to identify different ship activities. Duan et al. (2022) used a vector model incorporating ship-measured speed, wind speed, wave height, and water flow velocity to correct ship speed. The

results indicated that when the average wind speed, wave height, and ocean current were relatively high, specifically when the average wind speed exceeded 5.5 m s$^{-1}$, the average corrected speed only accounted for 5.88 % to 7.69 % of the measured speed, meaning that the impact on the engine workload of the ship was also minimal. In order to minimize the impact of meteorological conditions on the test results, we picked calm weather (with zephyrs and gentle waves) and appropriate temperature to carry out ship experiments as presented in Table S1 of SI. In conclusion, similar to some previous researches regarding NO$_x$ emitted from ships (Wang et al., 2019; Jiang et al., 2019; Zhang et al., 2018; Zhao et al., 2020), we did not take into account the influence of meteorological conditions during sampling, considering its minimal impact and difficulty in quantification. The final results also indicated that the operational status of the ship was not the most significant factor affecting the emission of $\delta^{15}$N–NO$_x$ values from ships.

Since our sampling was conducted during the actual operation of the ship, it is difficult to achieve identical conditions for all factors except for meteorological variables in two separate and non-consecutive sampling processes, even under the same operating condition on the same ship. Therefore, it is difficult to demonstrate that there are no systematic differences between the sampling processes in summer and winter. Subsequent studies on ship emissions can consider investigating these differences in a laboratory setting.

We have included the necessary explanation and relevant references regarding this in the revised manuscript: "Meteorological conditions primarily affect ship emissions by influencing the operational mode of the ship, and this impact is relatively small and difficult to quantify (Huang et al., 2018; Duan et al., 2022; Zhao et al., 2020). Considering the unpredictable weather circumstances, we attempted to pick calm weather (with zephyrs and gentle waves) and appropriate temperature to carry out ship experiments. As presented in Table S1 of the Supporting Information (SI), the temperature, wind speed and relative humidity ranged from 1 to 27°C, from 2.8 to 5.1 m s$^{-1}$, and from 49 to 68 % during the observation period, respectively, which were obtained from the local weather station established by the China Meteorological Administration (Wang et al., 2019). Consequently, similar to some previous researches regarding NO$_x$ emitted from ships (Wang et al., 2019; Jiang et al., 2019; Zhang et al., 2018; Zhao et al., 2020), the influence of meteorological conditions during sampling was not taken into account in this study." Page 4, Line 121–131

(2) Based on the analysis presented in (1), we believe that weather changes are not the primary cause of internal variability within the same operating condition. For further explanations about the reason for large variation of $\delta^{15}$N–NO$_x$ values within the same ship emissions under the same mode, please refer to the response to the third question in the "Results and discussion" section.

On line 145–146 background and blank samples are collected but the data from these samples is not discussed in the results and should be. On Line 160 it is stated that the average blank concentration was ~1.15 % — is this a lab blank? A field blank? Or an N$_2$O blank associated with the denitrifier method? How much average change results from correcting for the blank overall?

**Response and Revisions:** We agree with the reviewer that a fuller discussion of the blank samples is indeed warranted. First, to make the expression clearer, we supplemented the number of background blank samples in Section 2.1: "Additionally, samples for background blank were collected during the sampling period of each ship (3 samples per ship, totaling 45 samples) to quantify background $NO_3^-$ concentrations and to correct for isotope blanks." Page 6, Line 169–171

Next, the calibration of the $N_2O$ blank associated with the bacterial denitrifier procedure using the two-point correction method was added to Section 2.2: "Finally, the $\delta^{15}N$ of the produced $N_2O$ was analyzed by an isotope ratio mass spectrometer (MAT253, Thermo Fisher Scientific, Waltham, MA) and the $\delta^{15}N$ values were reported in parts per thousand relative to the international standards (IAEA–NO–3, USGS32, USGS34, and USGS35) (Bohlke et al., 2003):

$$\delta^{15}N = \left[ \frac{\left( ^{15}N/^{14}N \right)_{sample}}{\left( ^{15}N/^{14}N \right)_{standard}} - 1 \right] \times 1000 \qquad (1)"$$

Page 7, Line 181–185

The "1.15 ± 2.02 %" in line 160 of the previous manuscript refers to background blank samples related to laboratory and field sampling, and further discussion has been added in the revised manuscript: "The average $NO_3^-$ concentration of background blank samples from each ship was 45.33–682.50 ug N/L, accounting for 1.15 ± 2.02 % of the regular samples. This includes the background value of $NO_3^-$ in the absorption solution, and the $NO_3^-$ converted from $NO_x$ captured from ambient air during the collection time of 20 minutes. Therefore, the final $NO_3^-$ concentrations for samples from each vessel were recalculated by subtracting the average blank value during sampling. The average $\delta^{15}N$ values related to the background blank for each ship ranged from −3.02 ‰ to 11.34 ‰, and the $\delta^{15}N$ value for each sample was redetermined by mass balance (Fibiger et al., 2014), leading to an average variation in $\delta^{15}N$ values ranging from 1.12 % to 4.87 %:

$$\delta^{15}N = \frac{\delta^{15}N_{total}[NO_3^-]_{total} - \delta^{15}N_{blank}[NO_3^-]_{blank}}{[NO_3^-]_{total} - [NO_3^-]_{blank}} \qquad (2)$$

where $\delta^{15}N_{total}$ and $\delta^{15}N_{blank}$ are the $\delta^{15}N$ values (%) of samples and blanks of the ship, respectively; $[NO_3^-]_{total}$ and $[NO_3^-]_{blank}$ are $NO_3^-$ concentrations (ug N/L) of samples and blanks of the ship, respectively." Page 7, Line 186–197

Equation 2: How was this equation determined? Is it determined empirically? (i.e. based on data in this study). What is LF to the power of 3? And what units are used for each variable in this equation?

**Response and Revisions:** We apologize for the unclear description of this equation and misrepresentation of LF. LF (%) is the load factor of the main engine (ME) of the ship under different operating conditions, and can be calculated from the actual sailing speed (AS, knot) and the maximum design speed (MS, knot) of the ship (Chen et al., 2017):

$$LF = \left( \frac{AS}{MS} \right)^3$$

It was capped to 1.0 in the case the calculated value exceeded 100%. Therefore, the weighted value

of $\delta^{15}$N emitted from ME and the auxiliary engine (AE) should be calculated as:

$$\delta^{15}N=\frac{0.22 \times P_{ME} \times \delta^{15}N_{AE}+LF \times P_{ME} \times \delta^{15}N_{ME}}{0.22 \times P_{ME}+LF \times P_{ME}}$$

where $P_{ME}$ represent the rated power of ME, $0.22 \times P_{ME}$ is the AE power and $LF \times P_{ME}$ is the actual ME power of the vessel under various operating conditions. The formula is finally organized as:

$$\delta^{15}N=\frac{0.22 \times \delta^{15}N_{AE}+LF \times \delta^{15}N_{ME}}{0.22+LF}$$

Accordingly, we have modified this formula in the revised manuscript: "Since a ship's ME and AE work simultaneously for most times, the emission powers of the two were used for weighted calculations, and thus the actual $\delta^{15}$N–NO$_x$ values emitted by ships under each operating condition could be obtained as follows:

$$\delta^{15}N=\frac{0.22 \times \delta^{15}N_{AE}+LF \times \delta^{15}N_{ME}}{0.22+LF} \tag{3}$$

where $\delta^{15}N_{AE}$ and $\delta^{15}N_{ME}$ are the $\delta^{15}$N–NO$_x$ values (%) emitted by AE and ME of the ship, respectively; LF (%) is the load factor of ME under different operating conditions of the ship and can be calculated from the actual sailing speed (AS, knot) and the maximum design speed (MS, knot) of the ship (Chen et al., 2017):

$$LF=\left(\frac{AS}{MS}\right)^3 \tag{4}$$

It was capped to 1.0 in the case the calculated value exceeded 100%." Page 7–8, Line 204–213

**Results and discussion:**

The reported range of measurement is −35.8 per mil to 2.0 per mil. Are these the initial data or have these already been weighted based on Eq. 2? In order to re-calculate these values, the AE and ME fractions also need to be reported. Can this be included in a table in the supplement?

**Response and Revisions:** We appreciate the questions raised by the reviewer regarding this. The $\delta^{15}$N values reported in the results are all weighted data based on Eq. 2 (Eq. 3 in the revised manuscript) and these values do not need to be recalculated.

Suggest bringing Figure S7 into the main manuscript in Section 3.1. Also, consider formatting the symbols to be colored by whether the ship was operating under Tier I, Tier II or Tier III emissions regulations. It would be helpful to get a sense of the variability in concentration with the samples and this is not currently reported in the main manuscript.

**Response and Revisions:** Following the reviewer's opinion, we have supplemented Figure S7 in the revised manuscript as Figure 4 and used different symbols to represent samples that meet different ship emission regulations. Please refer to the updated Figure 4 presented below.

[Figure]

**Figure 4.** The negative logarithmic relationship between $\delta^{15}N–NO_x$ values and $NO_x$ concentration emitted from ships.

The variability in $NO_x$ concentrations emitted from ships under different emission standards was reported on page 14, line 411–413: "Meanwhile, compared to the $\delta^{15}N–NO_x$ values which progressively increased with the tighter policing, the corresponding $NO_x$ concentrations gradually decreased and were $239 \pm 93.0$ ppm, $169 \pm 156$ ppm, and $122 \pm 91.6$ ppm under Tier I, II and III, respectively, as presented in Table 3." In addition, we also summarized the information of each stage in Table3, including the measured $NO_x$ concentrations (ppm), and it has been supplemented in the revised manuscript.

This section should also include more discussion of the variability within an observed group – i.e., why is there such large variation within the same ship emissions under hoteling condition?

**Response and Revisions:** Under different operating conditions, ships have different speeds and loads, and factors such as fuel consumption rate, combustion temperature and time will change, which in turn affects the emission of $NO_x$ from ships. According to previous research by Chen et al. (2016), $NO_x$ samples in our study were grouped based on three operating modes on the basis of the actual speed of each ship: cruising mode (> 8 knots, ship operating at higher speed), maneuvering mode (1–8 knots, ship operating at lower speed when approaching berths or anchorages) and hoteling mode (< 1 knot, ship at berth or anchored). As shown in Table S2, significant differences in emissions under the same operating mode on the same ship were more often observed during hoteling and cruising conditions.

The hoteling mode typically occurs when a vessel is not departing from the port and not engaged in normal navigation operations, such as during cargo loading/unloading or while anchored awaiting further instructions. There are variations in emissions due to changes in the usage requirements of

different onboard equipment in the hoteling mode. Cooper (2003) found that for approximately 5 min after arrival at the quayside, and approximately 15 min before departure, the power requirement for ships studied increased to 40–56 % of the total installed AE power when bow and stern thrusters used for manoeuvring the ship were engaged. Additionally, cargo pumps used during the cargo handling process on bulk carriers and the refrigeration equipment for storing the catch on fishing vessels may lead to a significant increase in power demand during the hoteling mode. In our study, the state in which the vessel operated at a higher speed (> 8 knots) was defined as the cruising mode. This mode exhibited a wide range of variation in ship speed. Moreover, ships often operate in cruise mode when navigating in open seas far from the coast and are more likely to encounter larger waves and swells. As a result, the engine load of a ship in cruise mode is more susceptible to fluctuations and changes compared to other operating modes, and consequently lead to variations in the $NO_x$ measurement of exhaust samples collected during cruising conditions.

Therefore, we have explained the reason for large variation of $\delta^{15}N–NO_x$ values within the same ship emissions under the same mode in the revised manuscript: "There can be large variation in $\delta^{15}N–NO_x$ values emitted from the same vessel under the same operating condition as shown in Table S2, which was often observed during hoteling and cruising mode. This could be attributed to changes in the usage requirements of different onboard equipment in hoteling mode, as well as the differences in engine load concerning the substantial variations in vessel speed during cruising mode (Cooper, 2003; Huang et al., 2018). A more detailed interpretation can be found in Text S1 of SI." Page 9, Line 250–255

Is the distance from shore important at all for variation in $d^{15}N–NO_x$?

**Response and Revisions:** In our study, statistical analysis methods and machine learning techniques indicated that the ship-emitted $\delta^{15}N–NO_x$ values were mainly influenced by the adoption of $NO_x$ emission control technologies (e.g., the selective catalytic reduction system), the ship category and fuel type, rather than the distance from shore. The impact of the distance from shore during navigation is primarily related to the operational conditions of the vessel, which is one of the indicators discussed in our study. When ships navigate in coastal areas that are close to shore, they often operate in maneuvering and hoteling modes, resulting in certain variations in $\delta^{15}N–NO_x$ values.

Line 224: This last sentence does not follow from the evidence presented. The comparison suggests that it is necessary to know the types of engine in order to accurately apportion sources or to treat the mix of engines with respect to the fact that they each have different values associated with their $NO_x$ emissions.

**Response and Revisions:** We are very sorry for the unclear expression in the previous manuscript. In order to make the conclusion clearer and strengthen its logical connection with the evidence presented, we have rephrased this section as follows: "The significant difference observed highlights the importance of addressing the data gap in measuring $\delta^{15}N–NO_x$ emissions from ships for accurately apportioning atmospheric $NO_x$ sources in coastal zones, especially in some port areas

with high ship activity. Further discussion is needed to understand the reasons behind the noticeable difference and to better utilize the measured values of ship-emitted $\delta^{15}N–NO_x$, including the variations in $\delta^{15}N–NO_x$ values and their primary influencing factors." Page 10, Line 288–293

Furthermore, be sure it is appropriate to directly compares values here. Some methods may only collect NO2 while the methodology used here collects both NO and NO2.

**Response and Revisions:** The $\delta^{15}N–NO_x$ values emitted from vehicles in Figure 1 were reported in previous studies of Walters et al, where $NO_x$ in the exhaust were collected, rather than individual NO or $NO_2$ (Walters et al., 2015a; Walters et al., 2015b). Please refer to Table S3 in the SI for more sampling details. Thus, it is appropriate to directly compares the $\delta^{15}N–NO_x$ values displayed here.

(1) Starting page 10, the discussion of Figure S4 and the conclusions drawn from this analysis are a bit unclear/difficult to follow. More of the discussion and interpretation from Text S1 should be included here. (2) Also consider including Figure S4. And parts of Fig S4 are cutoff and unreadable, which also adds to the confusion. (3) (Also note that on line 276, the stages are called I, II, III etc. instead of 1, 2, 3… this may confuse readers bc of the Tier # system. (4) Also, Fig S4 does not show Stage 3 – why not?).

**Response and Revisions:** Due to this comment containing multiple sub-questions, we have divided it according to the meaning of questions (1) to (4). Therefore, we will provide answers to these questions as follows.

(1) Based on the reviewer's suggestion, we have included additional discussions from Text S1 of SI into the main text to provide a clearer explanation of the conditional inference trees (CIT) results: "The CIT analysis provided a more intuitive result as indicated in Fig. 3. It was found that the emission regulation met by ships was the most important splitting factor of the root node (node 1) and the second terminal node. Samples collected from ships prior to the implementation of IMO Tier I (stage 1) and ships implementing Tier I–III (stage 2–4) were separated to terminal node 9 ($-33.8 \pm 1.83$ ‰) and node 2 ($-16.8 \pm 9.50$ ‰), respectively, indicating a strong impact of implementing Annex V to the MARPOL 73/78 on these $\delta^{15}N–NO_x$ values. Then the emission regulation was again used as the next splitting factor to separate $\delta^{15}N–NO_x$ values emitted from ships meeting Tier III (stage 4) from node 2 as node 3 ($-7.93 \pm 5.33$ ‰). The $\delta^{15}N–NO_x$ values emitted from ships meeting Tier I & II (stage 2 and 3) as node 4 ($-18.3 \pm 9.25$ ‰) were subsequently divided by the ship category to node 5 (passenger ship and research ship, $-11.3 \pm 1.67$ ‰) and node 6 (cargo ship and fishing boat, $-21.0 \pm 9.48$ ‰). Finally, fuel type was the last splitting variable, separating samples taken when the ships used diesel (node 7, $-22.7 \pm 9.25$ ‰) and residual oil (node 8, $-11.2 \pm 1.38$ ‰) as fuel. The splitting process suggests that the emission regulation has a greater influence on $\delta^{15}N–NO_x$ values than the ship category." Page 11, Line 330–342

(2) We feel very sorry for the problem with the presentation of CIT results in Figure S4 and have made changes accordingly:

[Figure]

**Figure 3.** Conditional inference trees (CIT) for the $\delta^{15}N$–$NO_x$ values emitted from ships. For each inner node, the $p$ values are given and the range of $\delta^{15}N$–$NO_x$ values is displayed for each terminal node.

Figure S4 has also been supplemented into the revised manuscript as Figure 3.

(3) Thanks for the reminder from the reviewer. We have changed the numbering of the 4 emissions regulation stages from "I, II, III, IV" to "1, 2, 3, 4" (Page 12, Line 359, 365 and 366 in the revised manuscript).

(4) The third stage of the emission regulation is not represented in Figure S4 (Figure 3 in the revised manuscript) because the most significant factor influencing the $\delta^{15}N$–$NO_x$ values emitted by ships complying with Tier I (stage 2) and Tier II (stage 3) has shifted from the emission regulation to the vessel category. The vessel category further divided the $\delta^{15}N$–$NO_x$ values emitted by passenger ship & research ship (node 5), and cargo ship & fishing boat (node 6), which had significantly difference ($p$=0.001).

Figure 2/Line 305: the x-axis does not really plot emission regulation. It's really ship category, correct? I understand that different regulations apply to different types of ships, but the emission regulations are not actually stated on this plot. Also, what does the large x-range mean on the plots here? For instance, the passenger ship shows a significant bellowing out in the horizontal direction (as does the research ship results), but there is no quantity on the x-axis, so what does this range signify here?

**Response and Revisions:** We sincerely apologize for the incorrect categorization in Figure 2 in the previous manuscript. Following the reviewer's opinion, we have modified it to correctly display $\delta^{15}N$ values in $NO_x$ emitted from ships under different emission regulations established by the IMO.

Please refer to the updated figure presented below.

[Figure]

**Figure 2.** $\delta^{15}$N–NO$_x$ values emitted from ships under different emission regulations established by the IMO. (red square, mean; center line, median; box limits, upper and lower quartiles; whiskers, 1.5 times interquartile range; points, outliers (pink, cruising; green, hoteling; blue, maneuvering); outer line, distribution of data). Mean ± standard deviation of $\delta^{15}$N–NO$_x$ values of each group is marked on the bottom of the panel. The difference and $p$ values indicating the distinction between two groups are marked on the upper of the panel (the Mann–Whitney U test).

In this study, the violin plot was used to display ship-emitted $\delta^{15}$N–NO$_x$ values under different emission regulations established by the IMO. A violin plot is a type of chart used to visualize the distribution and density estimation of data. It combines features of a box plot and a kernel density plot to provide a visual representation of the median, quartiles, and overall shape of the data. In a violin plot, each "violin" represents a data variable or category. The width of the violin corresponds to the density of the data, with a wider violin indicating a higher data density. Therefore, in the above figure, the large x-range of the stage "Before Tier I" represents the most densely distributed $\delta^{15}$N–NO$_x$ values (± 1.83 ‰) emitted from ships not subject to NO$_x$ emission restrictions.

Section 3.3: This study should put forward recommendations on what values should be used in current studies for ship emissions. How sensitive is the d$^{15}$N from ship emissions to regional differences in ship fleets? Are there regional different in ship fleets? For instance, perhaps it could be calculated what regional values are so coastal studies in different locations could use those values. Or make a recommendation on how to calculate a value for your specific study region/study conditions.

**Response and Revisions:** We appreciate the reviewer's professional recommendations. This study collected the age distribution of ships larger than 300 gross tonnage (GT) in the international merchant fleet during 2001 and 2021 to assess the temporal variation in $\delta^{15}$N–NO$_x$ emitted from

ships by developing a mass-weighted model (Yang and Zhou, 2002; Zhou et al., 2004; Meng et al., 2005; Meng and Huang, 2006; Meng et al., 2007; Qi et al., 2008; Qin et al., 2009; Qi et al., 2010; Li et al., 2011; Qin et al., 2012; Qi et al., 2013; Li et al., 2014; Qin et al., 2015; Li et al., 2016; Qin et al., 2017, 2018; Shen and Qi, 2019; Qi et al., 2020; Liu et al., 2021; Wei et al., 2022). The calculated results only involved the age distribution and emission reduction level of the international merchant fleet, without considering regional variations. Therefore, if specific regional results for ship-emitted $\delta^{15}$N–NO$_x$ are required, weighted calculations (as shown in Equation 5 in the revised manuscript) should be performed using the age distribution of ships within that region. In other words, the spatial variation of the predicted results depends on the age distribution of ships in different regions. We have made a recommendation on how to calculate a value for the specific study region/study conditions: "In addition, given that the calculated results only involved the age distribution and emission reduction level of the international merchant fleet, the subsequent process of using $\delta^{15}$N to evaluate the contribution of ship emissions to atmospheric NO$_x$ can be combined with the actual situation of ships in the study area to select more appropriate $\delta^{15}$N–NO$_x$ values to acquire a more accurate ship emission contribution and reduce the uncertainty in NO$_x$ source apportionment." Page 17, Line 498–503

The manuscript should also include a final conclusions section.

**Response and Revisions:** We appreciate this suggestion from the reviewer, and have supplemented the Section 4, "Summary and conclusions". Please refer to the revised manuscript for further details.

**Minor technical comments:**

Line 57–58: The last sentence doesn't follow and then is repeated in the next paragraph. Suggesting removing sentence starting with "The rapid development…"

**Response and Revisions:** After considering the reviewer's suggestion, we have added a summarizing transitional sentence before the last sentence in the revised manuscript to enhance the overall coherence of the logic: "The current world merchant fleet thus comprises more newly built vessels that have benefited from the implementation of emission reduction technologies." Page 2–3, Line 64–65

Line 72: replace 'continuous updates', i.e., "These studies of ship emissions and their environmental impacts…"

**Response and Revisions:** As recommended by the reviewer, we have made the corresponding modifications. Page 3, Line 83–84

Line 74: replace 'some other' with 'additional'

**Response and Revisions:** In accordance with the suggestion of the reviewer, inappropriate expressions have been replaced. Page 3, Line 86

Line 77: please define delta units here d$^{15}$N does simply equal the ratio of $^{15}$N/$^{14}$N

**Response and Revisions:** Following the reviewer's opinion, we have provided the unit for $\delta^{15}N$ in this sentence: "NO$_x$ released into the atmosphere principally oxidizes to nitrate (NO$_3^-$) and nitric acid (HNO$_3$) and their nitrogen stable isotope composition ($\delta^{15}N$, i.e. $^{15}N/^{14}N$, expressed in ‰) is a powerful method to apportion NO$_x$ sources due to the significant differences in $\delta^{15}N$ values of NO$_x$ ($\delta^{15}N$–NO$_x$) from different sources (Walters et al., 2015a; Walters et al., 2015b; Zong et al., 2020a)." Page 3, Line 88–91 In addition, more reports about the unit for stable isotope ratios has been added in the revised Section 2.2 titled "Chemical and isotopic analysis" (page 7, line 181–185)."Finally, the $\delta^{15}N$ of the produced N$_2$O was analyzed by an isotope ratio mass spectrometer (MAT253, Thermo Fisher Scientific, Waltham, MA) and the $\delta^{15}N$ values were reported in parts per thousand relative to the international standards (IAEA–NO–3, USGS32, USGS34, and USGS35) (Bohlke et al., 2003):

$$\delta^{15}N=\left[\frac{\left(^{15}N/^{14}N\right)_{sample}}{\left(^{15}N/^{14}N\right)_{standard}}-1\right]\times1000 \qquad\qquad (1)"$$

Line 81: change 'emitted from' to 'characterized for'

**Response and Revisions:** As suggested by the reviewer, we have rephrased this expression. Page 3, Line 93

Line 98: change 'researches' to 'research'

**Response and Revisions:** After considering the reviewer's suggestion, we have rephrased this sentence: "The number of ships per category applied for NO$_x$ sample collection was determined based on previous reports that cargo ships accounted for more than 50 % of all NO$_x$ emissions from ships in China in 2014 and the fuel consumed by fishing boats accounted for 40 % of all ship fuel use in China in 2011 (Chen et al., 2017; Zhang et al., 2018; Zong et al., 2017)." Page 4, Line 110–113

Line 114: suggest changing 'emitted d$^{15}$N–NO$_x$' values to 'emitted NO$_x$'.

**Response and Revisions:** We have made the revisions based on the suggestion of the reviewer (page 6, line 135).

Line 117: this sentence is awkward. Suggest rephrasing to: "NO$_x$ emitted from AE were also collected, but the boiler was not sampled since these make a small contribution to NO$_x$ emissions…"

**Response and Revisions:** As recommended by the reviewer, corrections have been made to the grammar mistakes and inappropriate expressions in that sentence. "If the ship also has auxiliary engines (AE) and boilers, NO$_x$ emitted from AE were also collected, but boilers were not sampled since these make a small contribution to NO$_x$ emissions, less than 5% compared to those from ME and AE (Wan et al., 2020; Shi et al., 2020; Chen et al., 2017)." Page 6, Line 136–139

Line 149: Since for all sample redundant KMnO4 was removed this line appears awkward. Please rephrase.

**Response and Revisions:** We feel very sorry for the grammar mistake in that sentence and

corrections have been made: "The $NO_3^-$ concentration in the samples, where redundant $KMnO_4$ was removed, was quantified by standard colorimetric absorbance techniques (AutoAnalyzer 3, SEAL Analytical Ltd.) and the detection limit was 5 ng $mL^{-1}$." Page 7, Line 173–175

Line 153: what is an "absorption solution"? I think it meant the sample that was collected as nitrate? Replace this phrase.

**Response and Revisions:** Actually, "the absorption solution" refers to the solution in the gas-washing bottle during sampling to convert $NO_x$ into $NO_3^-$, containing 0.25 mol $L^{-1}$ potassium permanganate ($KMnO_4$) and 0.50 mol $L^{-1}$ sodium hydroxide (NaOH) (page 6, line 143). However, the intended meaning here is "the sample that was collected as $NO_3^-$" as suggested by the reviewer, thus we have made the necessary modifications to ensure a more accurate expression: "In short, the collected $NO_3^-$ solution containing 20 nmol N was put into the 20 mL headspace bottle and then 2 mL concentrated bacterial solution (helium-purged at 30 mL $min^{-1}$ for 4 h to alleviate the background interference) was added to convert $NO_3^-$ to nitrous oxide ($N_2O$). *P. aureofaciens* was selected as the experimental strain, which lacks the $N_2O$ reductase enzyme." Page 7, Line 177–180

Line 198: add '$NO_x$' so it reads – "…and $NO_x$ produced thermally by internal combustion engines…". Also should probably add the Snape paper that is cited in Walter's et al here or just say "and references therein".

**Response and Revisions:** Following the reviewer's opinion, "$NO_x$" has been added before the phrase "produced thermally by internal combustion engines tends to be..." and the Snape paper cited in Walter's et al has been supplemented in the list of cited references: "and $NO_x$ produced thermally by internal combustion engines tends to be depleted in $^{15}N$ abundance due to the kinetic isotope effect associated with the thermal decomposition of the strong triple bond of $N_2$ (Ti et al., 2021; Walters et al., 2015a; Walters et al., 2015b; Zong et al., 2020a; Snape et al., 2003)." Page 9, Line 262–265

Line 247: can it be mentioned what some "other indicators" are that have been used (as examples)?

**Response and Revisions:** We apologize for the lack of clarity in the previous manuscript. The term "other indicators" here refers to the ship category, type of fuel used, and the actual operational status of the ship mentioned earlier in the text. To avoid misunderstandings, we have rephrased the reasons for selecting these influencing factors: "The reason for considering compliance with the emission regulation for ship engines as a criterion is that the IMO emission regulation is the most important measure to restrict $NO_x$ emissions from ships and once each standard was proposed by the IMO, newly constructed or significantly refurbished marine diesel engines must comply with its requirements. Therefore, the emission regulation that a ship complies with is actually related to its age. Besides, the category, fuel type, and actual operational status are often used to assess the variation in $\delta^{15}N–NO_x$ values from vehicles (Walters et al., 2015a; Walters et al., 2015b)." Page 10–11, Line 313–319

Line 279: What are GB I to GB VI? These are not previously defined.

**Response and Revisions:** GB I to GB VI are a series of national vehicle emission standards for China, established by the Ministry of Ecology and Environment of the People's Republic of China (http://www.mee.gov.cn/). These standards aim to limit the emissions of pollutants in vehicle exhaust, including carbon monoxide, non-methane hydrocarbons, nitrogen oxides, and particulate matter. They apply to the production, sale, and use of automobiles nationwide. Currently, China's national standards have been continuously improved from GB I (implemented in 1999) to GB VI (implemented in 2019), which require that the concentration of pollutants emitted by vehicles under specific conditions must not exceed the specified limits. We appreciate the reviewer's reminder and have added the definition of GB in the revised manuscript: "This is in accordance with the variation trend of $\delta^{15}N–NO_x$ values emitted from gasoline vehicles complying with the national vehicle emission standards GB I to GB VI in China, which were established by the Ministry of Ecology and Environment of the People's Republic of China (Zong et al., 2020a)." Page 12, Line 360–363

Line 282–283: What is the "great discrepancy"? What does "hard to analyze" mean? This sentence over should be rephrased for clarity.

**Response and Revisions:** In this paragraph, following the statement "The largest average growth rate of $\delta^{15}N–NO_x$ values emitted from ships between the three adjacent phases from 1 to 4 occurred between the implementation of IMO Tier II and III (stage 3–4, 54.1 %), followed by before and after the implementation of Tier I (stage 1–2, 36.4 %), and between Tier I and II (stage 2–3, 17.8 %)" (page 12, line 363–366), we explained the reasons for the differences in $\delta^{15}N–NO_x$ values emitted from ships between two adjacent regulatory stages in chronological order. The "great discrepancy" here refers to the average growth rate of $\delta^{15}N–NO_x$ values emitted from ships before and after the implementation of Tier I (stage 1–2, 36.4 %). After careful consideration, we believe that there is an issue with the statement "hard to analyze". We deeply apologize for it and have removed the sentence.

Given that fuel optimization technologies and precombustion control technologies are two main control methods to meet Tier I and Tier II, we explained the reasons why the implementation of Tier I and Tier II resulted in increased ship-emitted $\delta^{15}N–NO_x$ values, as well as the reason why the average growth rate of $\delta^{15}N–NO_x$ values between Tier I and Tier II is lower than that before and after the implementation of Tier I in the revised manuscript: "Since the implementation of IMO Tier I on January 1, 2000, fuel optimization technologies (fuel emulsification, fuel desulfurization and fuel additives) and pre-combustion control technologies (fuel injection strategy, water injection strategy, Miller cycle, two-stage turbocharging, and dual-fuel combustion strategy), as demonstrated in Table 3, have been applied and experienced rapid development. The former reduce the production of $NO_x$ by changing the composition of fuel or adding additives to improve the combustion process, while the latter primarily suppress the formation of $NO_x$ by modifying the internal structure of diesel engines or adjusting parameters (Ampah et al., 2021; Deng et al., 2021; Lion et al., 2020). The average growth rate of $\delta^{15}N–NO_x$ values between Tier I and Tier II (17.8 %) is lower than that before

and after the implementation of Tier I (36.4 %). This difference can be attributed to the fact that different emission standards have varying levels of impact on $NO_x$ emission reductions. Compared to the optimization and development of emission reduction technologies between Tier I and II, the implementation of Tier I fills the gap in ship engine and fuel emission reduction measures, which clearly has a greater impact on $NO_x$ emissions." Page 12–13, Line 366–379

Line 321: please make this clearer when stating the strength of correlation is "just within range" – does that mean the r values are similar?

**Response and Revisions:** The strength of correlation is "just within range" actually does not mean the $r$ values are similar. In the correlation analysis, the $r$ value refers to the Pearson correlation coefficient, which measures the degree of linear correlation between two variables. A negative r value indicates a negative correlation, and the larger the absolute value, the stronger the correlation. For the correlation coefficient of the negative logarithmic correlation between $\delta^{15}N–NO_x$ values and $NO_x$ concentrations, the $r$ value (−0.39) for ship emissions is greater than the $r$ value (−0.1) for vehicle emissions without $NO_x$ emission control technologies, and smaller than the $r$ value (−0.92) for vehicle emissions with $NO_x$ emission control technologies. Therefore, we have concluded that the strength of the correlation between $\delta^{15}N–NO_x$ values and $NO_x$ concentrations of ship emissions is just within the range of those reported in vehicle emissions with or without $NO_x$ emission control technologies (page 14, line 421–423).

Line 324: ulteriorly is not correct here – perhaps "clearly"?

**Response and Revisions:** Following the reviewer's opinion, we have replaced "ulteriorly" with "further" in the revised manuscript. Page 14, Line 424

Line 326: is "thermodynamic" here the same as "thermal" which was previously used in the text?

**Response and Revisions:** We greatly appreciate the meticulous evaluation from the reviewer. The "thermodynamic" here is the same meaning as "thermal" which was previously used in the text. To steer clear of any possible misunderstandings, we have replaced "thermodynamic" with "thermal" here in the revised manuscript. Page 15, Line 428

Line 339: suggest changing to: "The catalytic reduction system is a major reason for the differences…"

**Response and Revisions:** According to the reviewer's suggestion, we have substituted "is one of the vital reasons for…" with "is a major reason for…". Page 15, Line 441

Line 383: What is Imo, 2019?

**Response and Revisions:** "Imo, 2019" refers to the Annex VI of the MARPOL implemented by IMO. We apologize for the improper citation of the reference and rephrased the sentence as: "The EF values in the four stages are 9.8 g $(KWh)^{-1}$, 9.8 g $(KWh)^{-1}$, 7.7 g $(KWh)^{-1}$, and 1.96 g $(KWh)^{-1}$, as presented in Table 3 (https://dieselnet.com/standards/inter/imo.php#other)." Page 16, Line 482–483

**Reviewer 3 (Page 29–39)**

Summary: The authors present an interesting study on characterizing the nitrogen stable isotope composition ($\delta^{15}$N) of nitrogen oxides (NO$_x$) from ship emissions. This work represents an important contribution to the field because ship emissions are an important source of NO$_x$; however, the $\delta^{15}$N values of this emission source had been previously unknown. The authors have characterized the $\delta^{15}$N–NO$_x$ from ship emissions from several different types of ship types that are also categorized by different emission standards. Overall, the authors find that the $\delta^{15}$N–NO$_x$ from ship emissions is extremely variable, similar to direct tailpipe emissions from light-duty vehicles. There is a strong dependency on the $\delta^{15}$N–NO$_x$ from ships that depend largely on the emission regulation standard and categorization of the ships. The authors then model the expected change in $\delta^{15}$N–NO$_x$ from ships emission as a function of emission regulations, helping to constrain the $\delta^{15}$N–NO$_x$ emission signature. This manuscript presents novel measurements and is generally suitable for publication in ACP. My largest gripe with this work is that a lot of useful material and figures are reported in the Supplement, which is critical to understanding the work, and parts of the Supplement should be moved into the main text. My recommendation is publication after minor revisions. My specific comments are provided below.

**Response and Revisions:** We appreciate the enlightening suggestions from the reviewer. We have made the necessary adjustments and enhancements to the manuscript, particularly by moving some important content from the Supplementary Information (SI) to the main text. A thorough response is provided below for reference. Thanks again for these thoughtful insights.

**Comments:**

Line 21–23: You may also mention that $\delta^{15}$N–NO$_x$ from ship emissions could also be important for source apportionment of atmospheric nitrogen deposition in remote ocean regions.

**Response and Revisions:** Thanks for further complementing the importance of $\delta^{15}$N–NO$_x$ values. We have added the sentence into the revised manuscript: "In addition, $\delta^{15}$N–NO$_x$ values from ship emissions could also be important for source apportionment of atmospheric nitrogen deposition in remote ocean regions." Page 1, Line 23–24

Lines 31–34: Besides NO$_x$, it could also be useful for tracking its influence in atmospheric nitrate air quality and N deposition studies.

**Response and Revisions:** In light of the reviewer's advice, we have modified this sentence as: "These simulated $\delta^{15}$N–NO$_x$ values can be used to select suitable $\delta^{15}$N–NO$_x$ values for a more accurate assessment, including the contribution of ship-emitted exhaust to atmospheric NO$_x$ and its influence on atmospheric nitrate (NO$_3^-$) air quality and nitrogen deposition studies." Page 1, Line 35–37

Lines 104–105: Table S1 could be useful to include in the main text. I was left wondering about the emission regulation categorization of the various types of ships throughout the manuscript, and

having a table to reference this information would be extremely useful.

**Response and Revisions:** In response to the guidance from the reviewer, we have added Table S1 as Table 1 in the revised text. Moreover, a table which provides a summary of emission regulations for ship-emitted $NO_x$ at each stage is indeed extremely helpful in clarifying the significant impact of emission regulations on $\delta^{15}N–NO_x$ values emitted from ships. Therefore, we have summarized the information of each stage, including the effective date of the emission regulation, limits for $NO_x$ emission factors (g/kWh), $NO_x$ emission reduction technologies, and the measured $NO_x$ concentrations (ppm) and $\delta^{15}N–NO_x$ values (‰) in our study, in Table 3, and it has been supplemented in the revised manuscript.

**Table 3.** Summary of ship-emitted $NO_x$ at each stage of emission regulations.

| stage | effective date of the emission regulation | $NO_x$ emission limits (g (KWh)$^{-1}$) | | | $NO_x$ emission reduction technologies | $NO_x$ (ppm) | $\delta^{15}N–NO_x$ (‰) |
|---|---|---|---|---|---|---|---|
| | | $N^a <$ 130 | $130 \leq N$ < 2000 | $N \geq$ 2000 | | | |
| Before Tier I | | | | | | 96.6 ± 28.6 | −33.8 ± 1.83 |
| Tier I | 2000/01/01 | 17.0 | 45N$^{-0.2}$ | 9.8 | fuel optimization: fuel emulsification, fuel desulfurization, fuel additives; pre-combustion control technologies: fuel injection, water injection, Miller cycle, two-stage turbocharging, dual-fuel combustion | 239 ± 93.0 | −21.5 ± 6.67 |
| Tier II | 2011/01/01 | 14.4 | 44N$^{-0.23}$ | 7.7 | | 169 ± 156 | −17.8 ± 9.88 |
| Tier III (in NECAs$^b$) | 2016/01/01 | 3.4 | 9N$^{-0.2}$ | 1.96 | exhaust after-treatment system: exhaust gas recirculation (EGR), selective catalytic reduction (SCR), non-thermal plasma (NTP), seawater flue gas desulphurization (SWFGD) | 122 ± 91.6 | −8.12 ± 8.84 |

[a] N: rated speed (rpm)

[b] NECAs: the Nitrogen Oxide Emission Control Areas, mainly including the Baltic Sea area, the North American Emission Control Area, and the Caribbean Sea Emission Control Area.

Lines 110–112: Whether meteorological conditions would impact $\delta^{15}N–NO_x$ is unclear. Can the authors expand on this point or delete this sentence entirely?

**Response and Revisions:** Meteorological conditions primarily affect ship emissions by influencing the operational mode of the ship. For example, sailing with a tailwind or headwind can result in variations in the engine load, thereby altering the $\delta^{15}N–NO_x$ values from ship emissions. Quantifying the exact extent of these changes may pose challenges. Ship speed indicating workloads of ship engines is one of key factors to identify different ship activities. Duan et al. (2022) used a vector model incorporating ship-measured speed, wind speed, wave height, and water flow velocity to correct ship speed. The results indicated that when the average wind speed, wave height, and ocean current were relatively high, specifically when the average wind speed exceeded 5.5 m s$^{-1}$, the average corrected speed only accounted for 5.88 % to 7.69 % of the measured speed, meaning that the impact on the engine workload of the ship was also minimal.

In order to minimize the impact of meteorological conditions on the test results, we picked calm weather (with zephyrs and gentle waves) and appropriate temperature to carry out ship experiments as presented in Table S1 of SI. In conclusion, similar to some previous researches regarding $NO_x$ emitted from ships (Wang et al., 2019; Jiang et al., 2019; Zhang et al., 2018; Zhao et al., 2020), we did not take into account the influence of meteorological conditions during sampling, considering its minimal impact and difficulty in quantification. The final results also indicated that the operational status of the ship was not the most significant factor affecting the emission of $\delta^{15}N$–$NO_x$ values from ships.

We have included the necessary explanation and relevant references regarding this in the revised manuscript: "Meteorological conditions primarily affect ship emissions by influencing the operational mode of the ship, and this impact is relatively small and difficult to quantify (Huang et al., 2018; Duan et al., 2022; Zhao et al., 2020). Considering the unpredictable weather circumstances, we attempted to pick calm weather (with zephyrs and gentle waves) and appropriate temperature to carry out ship experiments. As presented in Table S1 of the Supporting Information (SI), the temperature, wind speed and relative humidity ranged from 1 to 27°C, from 2.8 to 5.1 m s$^{-1}$, and from 49 to 68 % during the observation period, respectively, which were obtained from the local weather station established by the China Meteorological Administration (Wang et al., 2019). Consequently, similar to some previous researches regarding $NO_x$ emitted from ships (Wang et al., 2019; Jiang et al., 2019; Zhang et al., 2018; Zhao et al., 2020), the influence of meteorological conditions during sampling was not taken into account in this study." Page 4, Line 121–131

Lines 115–117: Can the authors quantify the relative contribution of the boiler emissions to the main and auxiliary engines? Is it, for example, less than 5 %?

**Response and Revisions:** According to the previous researches about ship emissions from different engines, the $NO_x$ emissions from boilers are less than 5% compared to those from the main engines (ME) and auxiliary engines (AE). For example, Shi et al. (2020) explored the effectiveness of emission control areas (ECA) policies in reducing pollutant emissions from merchant ships in Shanghai port waters and found that the $NO_x$ emissions from boilers accounted for 2.40 % of the total $NO_x$ emissions from all engines of cargo ships. Based on port visiting records, Wan et al. (2020) estimated ship pollutant emissions in the Bohai Bay, the Yangtze River Delta, and the Pearl River Delta of China in 2018. The $NO_x$ emissions from boilers comprised only about 1 % of the total emissions from all engines, as shown in Figure C1 below.

[Figure]

**Figure C1.** Ship emissions of different engines in the research conducted by Wan et al. (2020).

Additionally, Chen et al. (2017) developed the first comprehensive ship emission inventory in China including ocean-going vessels, coastal vessels and river vessels, where boilers contributed merely 3.5 % of the $NO_x$ emissions because they are mostly used at berth or in the offshore area (Figure C2).

[Figure]

**Figure C2.** Shares of ship emissions classified by ship engine in the research conducted by Chen et al. (2017).

Consequently, we quantified the relative contribution of the boiler emissions to the main and auxiliary engines in the revised manuscript on the basis of the reviewer's advice: "$NO_x$ exhaust from the main engines (ME) of ships was collected. If the ship also has auxiliary engines (AE) and boilers, $NO_x$ emitted from AE were also collected, but boilers were not sampled since these make a small contribution to $NO_x$ emissions, less than 5% compared to those from ME and AE (Wan et al., 2020; Shi et al., 2020; Chen et al., 2017)." Page 6, Line 135–139

Lines 117–119: Reactive gases can collect on stainless steel. How do the authors expect the sampling apparatus to impact the isotope results? Also, was water formed/condensed on the sampling apparatus? Was there a relationship between collection time and the measured $\delta^{15}N–NO_x$?

**Response and Revisions:** Thanks for the reviewer's reminder. In this study, due to the fact that the

exhaust stacks of sampling ships are usually located at higher or lateral positions, a supportive pipe is required to guide the exhaust gases to a platform where sampling devices can be installed. Based on previous empirical studies on ship emission characteristics, such as researches by Zhao et al. (2020) and Liu et al. (2022), the combination of stainless steel bellows and Teflon tubes is most commonly used for ship sampling. Consequently, we also employ this method as it offers better operability and comparability.

Throughout the sampling process, to avoid the occurrence of $\delta^{15}N$ fractionation, each sampling process and analyzing process were conducted carefully and exactly. The connecting pipes used were stainless steel bellows (1.5 m or 3.0 m with an inner diameter of 40 mm) and Teflon tubes (1.5 m with an inner diameter of 12.77 mm); and the exhaust gas was pumped at a flow rate of 1.0 L min$^{-1}$ into the gas-washing bottle. Therefore, the time for which NO$_x$ were present in the pipe was less than 4 min. This was much lower than the average lifetime of NO in the air ($\sim$15216.3 s) (Massman, 1998); the fractional distillation could thus be ignored. We apologize for the previous calculation error and have made the necessary corrections in the revised manuscript: "The total length of the connecting pipe (stainless steel bellows and Teflon tubes) from the ship chimneys to gas-washing bottles was about 2 m or 4 m for different ships and NO$_x$ were accordingly present in the pipe for less than 4 min, which was significantly shorter than the normal airborne NO lifespan ($\sim$15216.3 s), meaning that fractional distillation could be ignored (Massman, 1998)." Page 6, Line 161–165

Each sample was typically ensured to have a sampling time of 20 minutes. This is because it is necessary to collect a sufficient amount of NO$_x$ to meet the detection limit of the bacterial denitrifier method. Additionally, an adequate collection time ensures that the concentration of NO$_3^-$ converted by NO$_x$ of the exhaust in the potassium permanganate (KMnO$_4$) reserve solution is higher than the background NO$_3^-$ concentration, minimizing uncertainties related to isotope blank correction. This method was proved in the previous studies (Fibiger et al., 2014; Zong et al., 2020a) to be effective at collecting 100 % ($\pm$5 %, 1$\sigma$) of the NO$_x$ and producing consistent isotope results under a wide variety of conditions. Therefore, the experimentally obtained $\delta^{15}N$ values are essentially stable as long as the sampling time is sufficient.

Lines 126–128: Can the collection time for each sample be provided in the Supplement?

**Response and Revisions:** We sincerely apologize for the unclear expression in the previous manuscript. In fact, each sample was typically ensured to have a sampling time of 20 minutes to meet the detection limit of the bacterial denitrifier method and to minimize uncertainties related to isotope blank correction. Accordingly, we rephrased this sentence in the revised manuscript: "Each sample was collected continuously for 20 min after 5 min of stable operation in each operating mode of the ship. An adequate sampling time is essential to ensure that sufficient amount of NO$_3^-$ was collected for conducting $\delta^{15}N$ measurement and minimizing uncertainties associated with isotope blank correction." Page 6, Line 149–152

Lines 149–151: What was the precision of the $NO_3^-$ concentration measurement?

**Response and Revisions:** The analytical precision of $NO_3^-$ concentrations was less than 1.8 % as determined by the replicates in this study, which can be found at the end of Section 2.2 "Chemical and isotopic analysis" on page 7, line 197.

Lines 153–155: You may mention that this particular strain of bacteria lacks the $N_2O$ reductase enzyme, which is critical for the analysis.

**Response and Revisions:** In agreement with the reviewer's feedback, we have rephrased the sentence as: "In short, the collected $NO_3^-$ solution containing 20 nmol N was put into the 20 mL headspace bottle and then 2 mL concentrated bacterial solution (helium-purged at 30 mL min$^{-1}$ for 4 h to alleviate the background interference) was added to convert $NO_3^-$ to nitrous oxide ($N_2O$). *P. aureofaciens* was selected as the experimental strain, which lacks the $N_2O$ reductase enzyme." Page 7, Line 177–180

Lines 160–162: Can you provide the average blank concentration and the number of blank samples taken in addition to the blank fractional contribution? Additionally, what was the measured $\delta^{15}N$ of the blank?

**Response and Revisions:** We agree with the reviewer that a fuller discussion of the blank samples is indeed warranted. First, we supplemented the number of background blank samples in Section 2.1: "Additionally, samples for background blank were collected during the sampling period of each ship (3 samples per ship, totaling 45 samples) to quantify background $NO_3^-$ concentrations and to correct for isotope blanks." Page 6, Line 169–171

Next, the calibration of the $N_2O$ blank associated with the bacterial denitrifier procedure using the two-point correction method was added to Section 2.2: "Finally, the $\delta^{15}N$ of the produced $N_2O$ was analyzed by an isotope ratio mass spectrometer (MAT253, Thermo Fisher Scientific, Waltham, MA) and the $\delta^{15}N$ values were reported in parts per thousand relative to the international standards (IAEA–NO–3, USGS32, USGS34, and USGS35) (Bohlke et al., 2003):

$$\delta^{15}N = \left[ \frac{\left(^{15}N/^{14}N\right)_{sample}}{\left(^{15}N/^{14}N\right)_{standard}} - 1 \right] \times 1000 \qquad (1)"$$

Page 7, Line 181–185

The "1.15 ± 2.02 %" in line 160 of the previous manuscript refers to background blank samples related to laboratory and field sampling. Further discussion, including the average blank values of $NO_3^-$ concentration and $\delta^{15}N$ for each ship, has been added in the revised manuscript: "The average $NO_3^-$ concentration of background blank samples from each ship was 45.33–682.50 ug N/L, accounting for 1.15 ± 2.02 % of the regular samples. This includes the background value of $NO_3^-$ in the absorption solution, and the $NO_3^-$ converted from $NO_x$ captured from ambient air during the collection time of 20 minutes. Therefore, the final $NO_3^-$ concentrations for samples from each vessel were recalculated by subtracting the average blank value during sampling. The average $\delta^{15}N$ values

related to the background blank for each ship ranged from −3.02 ‰ to 11.34 ‰, and the $\delta^{15}$N value for each sample was redetermined by mass balance (Fibiger et al., 2014), leading to an average variation in $\delta^{15}$N values ranging from 1.12 % to 4.87 %:

$$\delta^{15}N = \frac{\delta^{15}N_{total}[NO_3^-]_{total} - \delta^{15}N_{blank}[NO_3^-]_{blank}}{[NO_3^-]_{total} - [NO_3^-]_{blank}} \tag{2}$$

where $\delta^{15}N_{total}$ and $\delta^{15}N_{blank}$ are the $\delta^{15}$N values (%) of samples and blanks of the ship, respectively; $[NO_3^-]_{total}$ and $[NO_3^-]_{blank}$ are $NO_3^-$ concentrations (ug N/L) of samples and blanks of the ship, respectively." Page 7, Line 186–197

Line 174: This equation should be modified to more clearly indicate which molecule $\delta^{15}$N corresponds to.

**Response and Revisions:** We apologize for the unclear description of this equation and misrepresentation of LF. LF (%) is the load factor of ME of the ship under different operating conditions, and can be calculated from the actual sailing speed (AS, knot) and the maximum design speed (MS, knot) of the ship:

$$LF = \left(\frac{AS}{MS}\right)^3$$

It was capped to 1.0 in the case the calculated value exceeded 100%. Therefore, the weighted value of $\delta^{15}$N emitted from ME and AE should be calculated as:

$$\delta^{15}N = \frac{0.22 \times P_{ME} \times \delta^{15}N_{AE} + LF \times P_{ME} \times \delta^{15}N_{ME}}{0.22 \times P_{ME} + LF \times P_{ME}}$$

where $P_{ME}$ represent the rated power of ME, $0.22 \times P_{ME}$ is the AE power and $LF \times P_{ME}$ is the actual ME power of the vessel under various operating conditions. The formula is finally organized as:

$$\delta^{15}N = \frac{0.22 \times \delta^{15}N_{AE} + LF \times \delta^{15}N_{ME}}{0.22 + LF}$$

Accordingly, we have modified this formula in the revised manuscript: "Since a ship's ME and AE work simultaneously for most times, the emission powers of the two were used for weighted calculations, and thus the actual $\delta^{15}$N–NO$_x$ values emitted by ships under each operating condition could be obtained as follows:

$$\delta^{15}N = \frac{0.22 \times \delta^{15}N_{AE} + LF \times \delta^{15}N_{ME}}{0.22 + LF} \tag{3}$$

where $\delta^{15}N_{AE}$ and $\delta^{15}N_{ME}$ are the $\delta^{15}$N–NO$_x$ values (%) emitted by AE and ME of the ship, respectively; LF (%) is the load factor of ME under different operating conditions of the ship and can be calculated from the actual sailing speed (AS, knot) and the maximum design speed (MS, knot) of the ship (Chen et al., 2017):

$$LF = \left(\frac{AS}{MS}\right)^3 \tag{4}$$

It was capped to 1.0 in the case the calculated value exceeded 100%." Page 7–8, Line 204–213

Lines 178–187: These statistical tests are not common, and a description should be provided in the text, not the Supplement.

**Response and Revisions:** Consistent with the reviewer's viewpoint, we have incorporated the detailed descriptions of various statistical methods from the Text S1 of the previous SI into the main body of our paper as follows:

"Among the statistical analysis methods, the analysis of variance (ANOVA) and the Mann–Whitney U test were applied to support the factors influencing $\delta^{15}N$–$NO_x$ values from ships and further compare whether there was significant discrepancy between different classifications under the influence factor, respectively. ANOVA is a statistical method used to compare the means of two or more groups to determine if there is a significant difference. The core idea of ANOVA is to compare the differences between groups to the differences within groups. It calculates the ratio of between-group variation to within-group variation, known as the $F$ value, and then compare this $F$ value with the $F$ critical value corresponding to the given significance level to determine if the mean differences are statistically significant. The Mann–Whitney U test is widely used in various fields for evaluating differences between two groups by comparing whether the medians of two samples are the same (Mann and Whitney, 1947). The Mann–Whitney U test typically reports a $p$ value, which represents the probability of observing the current test statistic or a more extreme test statistic under the same null hypothesis conditions. If the $p$ value is less than the significance level, the null hypothesis that the medians of the two samples are the same can be rejected.

The conditional inference tree (CIT), random forest (RF), and the boosted regression tree (BRT) were implemented to quantitatively evaluate the impact degree of different factors on the variation in ship-emitted $\delta^{15}N$–$NO_x$ values. 75 % of the sample data were utilized to generate the prediction model, and the remaining data were utilized to evaluate the accuracy of the simulation results of the prediction model. The CIT is a non-parametric decision tree algorithm that recursively binary splits the dependent variable based on the values of correlations. It can handle features with different scales and selects features in an unbiased manner, as the feature and the best split point are determined after the feature selection (Hothorn et al., 2006). RF is an ensemble of regression trees originally used for classification. It evaluates the importance of candidate predictor variables by measuring the variance reduction in predictive accuracy before and after permuting the variables and combines the predictions of multiple trees to improve the overall model's performance (Strobl et al., 2007; Speybroeck, 2012). The BRT combines the strengths of regression trees and boosting, which is an adaptive method that combines many simple models to improve predictive performance (Elith et al., 2008). In the operation process, BRT randomly selects a subset of data multiple times to analyze the impact of predictor variables on the dependent variable and uses the remaining data to validate the fitting results. The output is the average of the generated regression trees. BRT is tolerant to covariance among predictors and non-normality, and it is less prone to overfitting, thus providing higher predictive accuracy for new data. These statistical analysis methods were conducted by R 4.1.3 software." Page 8, Line 215–246

Lines 190–191: It would be nice to reference a figure when discussing/presenting the data.

**Response and Revisions:** According to the reviewer's suggestion, we have referenced Figure 1 in the discussion here on ship-emitted $\delta^{15}N–NO_x$ values: "As illustrated in Fig. 1, the $\delta^{15}N–NO_x$ values emitted from ships sampled in this study were in the range of $-35.8$ ‰ to $2.04$ ‰, with a mean $\pm$ standard deviation of $-18.5 \pm 10.9$ ‰." Page 9, Line 249–250

Line 199– You should expand on the kinetic isotope effect mentioned in this line to the kinetic isotope effect associated with the thermal decomposition of the strong triple bond of $N_2$.

**Response and Revisions:** We have taken the reviewer's professional recommendations into account and made the corresponding changes (page 9, line 263–264).

Lines 251–252: The outcomes of the variance tests are interesting and dictate a lot of the discussion in the upcoming section. Thus, I would recommend including Table S5 in the main text.

**Response and Revisions:** We have incorporated the reviewer's suggestion and added Table S5 as Table 2 in the main text.

Figure 272–273: Figure 2 is mentioned to present the $\delta^{15}N–NO_x$ from ships under different emission regulations, but instead, Figure 2 shows the categorization of ship type. The $\delta^{15}N–NO_x$ by emission regulations is shown in Figure S2. Since these two factors are suggested to be the dominant driver of $\delta^{15}N–NO_x$ and are thoroughly discussed in this section, I recommend combining Figure 2 and S2 into one figure showing the $\delta^{15}N–NO_x$ relationship with both parameters.

**Response and Revisions:** We sincerely apologize for the incorrect categorization in Figure 2 in the previous manuscript. Following the reviewer's opinion, we have modified it to correctly display $\delta^{15}N$ values in $NO_x$ emitted from ships under different emission regulations established by the IMO. Please refer to the updated figure presented below.

[Figure]

**Figure 2.** $\delta^{15}N–NO_x$ values emitted from ships under different emission regulations established by the IMO. (red square, mean; center line, median; box limits, upper and lower quartiles; whiskers, 1.5 times interquartile range; points, outliers (pink, cruising; green, hoteling; blue, maneuvering); outer line,

distribution of data). Mean ± standard deviation of $\delta^{15}N-NO_x$ values of each group is marked on the bottom of the panel. The difference and $p$ values indicating the distinction between two groups are marked on the upper of the panel (the Mann–Whitney U test).

And Figure S2 (as Figure S3 in the revised SI) accurately illustrated the $\delta^{15}N-NO_x$ values emitted from ships grouped by different ship categories.

Lines 329–334: This enrichment factor, however, depends on how much $NO_x$ is reduced, and a shift in $\delta^{15}N$, which appears to have been applied to the measured data set, may not be appropriate.

**Response and Revisions:** We are grateful to the reviewer for the professional query. Walters et al. (2015a) studied the impact of catalytic reduction of $NO_x$ on $\delta^{15}N-NO_x$ and found that the $NO_x$ became enriched in $^{15}N$ relative to thermally produced NO, likely as a result of the equilibrium isotope effect between $N_2$ and $N^{14}O$ as well as the kinetic isotope effect associated with the diffusion and adsorption of $NO_x$ onto the catalytic converter surface. Furthermore, the observed $\delta^{15}N-NO_x$ and $NO_x$ concentration were used in a Rayleigh distillation model to determine the catalytic converter's net isotope effect, which includes diffusion, equilibrium, and kinetic effects:

$\delta^{15}N_f = \delta^{15}N_0 - \varepsilon \ln[NO_x]$

where $\delta^{15}N_f$ is the measured $\delta^{15}N-NO_x$, $\delta^{15}N_0$ is $\delta^{15}N-NO_x$ value that vehicle-emitted $NO_x$ reaches as $NO_x$ emissions approach zero, and $\varepsilon$ is the enrichment factor for $\delta^{15}N$ as the concentration of $NO_x$ decreases.

As presented in Figure C3, the enrichment factors concerning catalytic $NO_x$ reduction relative to the original thermal $NO_x$ for light- and heavy-duty diesel-powered engines were 6.1 ‰ and 8.7 ‰ respectively, while the enrichment factor for gasoline-powered engines is 5.2 ‰ (Walters et al., 2015a; Walters et al., 2015b). In conclusion, it is appropriate to correct the $\delta^{15}N-NO_x$ values from ships, diesel vehicles and gasoline vehicles not equipped with $NO_x$ catalytic reduction devices with these enrichment factors in our study.

[Figure]

(a)                (b)

**Figure C3.** $\delta^{15}N-NO_x$ (‰) as a function of collected $\ln(NO_x)$ (ppm). (a) The black data points represent samples collected from vehicles while in neutral and white data points represent exhaust samples collected from vehicles while driven; square points represent gasoline-powered engines, and circle points

represent light-duty diesel-powered engines (Walters et al., 2015a). (b) The points represent heavy-duty diesel-powered engines; linear fit is indicated by the red line, and the 95% confidence interval is shown in light red (Walters et al., 2015b).

Lines 359–402: I found this section interesting but was left with a few questions that I hope the authors can comment on – (1) will there be large spatial variability in the predicted $\delta^{15}$N–NO$_x$? (2) Also, how might the constrained $\delta^{15}$N–NO$_x$ ship values impact previous source appointment studies?

**Response and Revisions:** We appreciate the reviewer's interest in Section 3.3 and are willing to answer the questions raised by the reviewer.

(1) This study collected the age distribution of ships larger than 300 gross tonnage (GT) in the international merchant fleet during 2001 and 2021 to assess the temporal variation in $\delta^{15}$N–NO$_x$ emitted from ships by developing a mass-weighted model (Yang and Zhou, 2002; Zhou et al., 2004; Meng et al., 2005; Meng and Huang, 2006; Meng et al., 2007; Qi et al., 2008; Qin et al., 2009; Qi et al., 2010; Li et al., 2011; Qin et al., 2012; Qi et al., 2013; Li et al., 2014; Qin et al., 2015; Li et al., 2016; Qin et al., 2017, 2018; Shen and Qi, 2019; Qi et al., 2020; Liu et al., 2021; Wei et al., 2022). The calculated results only involved the age distribution and emission reduction level of the international merchant fleet, without considering regional variations. Therefore, if specific regional results for ship-emitted $\delta^{15}$N–NO$_x$ are required, weighted calculations (as shown in Equation 5 in the revised manuscript) should be performed using the age distribution of ships within that region. In other words, the spatial variation of the predicted results depends on the age distribution of ships in different regions.

(2) The considered sources in tracing sources of atmospheric NO$_x$ based on Bayesian models and $\delta^{15}$N–NO$_x$ values characterized for various sources were commonly coal combustion, vehicle exhaust, biomass burning and biological soil emissions (Luo et al., 2019; Song et al., 2020; Song et al., 2019; Zong et al., 2020b; Zhu et al., 2021; Yin et al., 2022). Due to the relatively close $\delta^{15}$N–NO$_x$ values between ship emissions and diesel vehicle emissions as shown in Table S3 of the revised SI, it may lead to an increase in the contribution of mobile sources in previous NO$_x$ source apportionment studies. Consequently, supplementing $\delta^{15}$N–NO$_x$ values from ships can greatly enhance the accuracy of atmospheric NO$_x$ source apportionment based on $\delta^{15}$N signals in offshore areas, especially in some ports with frequent ship activities.

**Technical Comment:**

TC1: Isotope deltas ($\delta$) are a quantity symbol and should be italicized throughout the manuscript.

TC2: The "$x$" in NO$_x$ should be italicized throughout the manuscript.

TC3: All units, including ‰, should be separate from the value by a space.

**Response and Revisions:** Thanks for the professional advice on paper writing. In the revised manuscript, we have corrected "$\delta$" and the subscript "$x$" in NO$_x$ to be in italics, and separated all units and values with spaces.

**Reference**

[revised manuscript text omitted]